# Continental shelves as a variable but increasing global sink for atmospheric carbon dioxide

Goulven G. Laruelle [1], Wei-Jun Cai [2], Xinping Hu[3], Nicolas Gruber [4], Fred T. Mackenzie[5] & Pierre Regnier[1]

It has been speculated that the partial pressure of carbon dioxide ($pCO_2$) in shelf waters may lag the rise in atmospheric $CO_2$. Here, we show that this is the case across many shelf regions, implying a tendency for enhanced shelf uptake of atmospheric $CO_2$. This result is based on analysis of long-term trends in the air–sea $pCO_2$ gradient ($\Delta pCO_2$) using a global surface ocean $pCO_2$ database spanning a period of up to 35 years. Using wintertime data only, we find that $\Delta pCO_2$ increased in 653 of the 825 0.5° cells for which a trend could be calculated, with 325 of these cells showing a significant increase in excess of +0.5 µatm yr$^{-1}$ ($p < 0.05$). Although noisier, the deseasonalized annual data suggest similar results. If this were a global trend, it would support the idea that shelves might have switched from a source to a sink of $CO_2$ during the last century.

[1] Department of Geoscience, Environment & Society, Université Libre de Bruxelles, Brussels, 1050, Belgium. [2] School of Marine Science and Policy, University of Delaware, Newark, DE 19716, USA. [3] Department of Physical and Environmental Sciences, Texas A&M University—Corpus Christi, Corpus Christi, TX 78412, USA. [4] Institute of Biogeochemistry and Pollutant Dynamics, ETH Zurich, 8092 Zurich, Switzerland. [5] Department of Oceanography, School of Ocean and Earth Science and Technology, University of Hawaii at Manoa, Honolulu, HI 96822, USA. Correspondence and requests for materials should be addressed to G.G.L. (email: goulven.gildas.laruelle@ulb.ac.be)

The atmospheric partial pressure of $CO_2$ ($pCO_{2,air}$) has been increasing at a rate of about 1.8 parts per million by volume (ppmv) per year in recent decades as a result of human activities such as burning of fossil fuel, deforestation, and cement production[1,2]. Although substantial regional and decadal variability has been observed[3,4], surface water $pCO_2$ levels tended to have followed more or less those of the atmosphere, particularly in the open ocean[5]. This tracking trend is best shown by the data collected at regular intervals at a few ocean time series stations, which by now cover more than 30 years[6]. The close atmospheric tracking of surface water $pCO_2$ is a consequence of the relatively long water residence time of the global surface ocean, with a time scale of more than a year[7], which is longer than an air–sea $CO_2$ exchange time scale of about 10 months[7]. However, it is not clear whether surface water $pCO_2$ on continental shelves, defined here as shallow regions with depths between 20 and 200 m that exclude the very nearshore areas (see "Methods" section for details), also track the atmospheric $pCO_2$ increase.

Our current understanding of the long-term trend in shelf $pCO_2$ is very limited because it largely relies on observations from a few time series only with records much shorter than those in the open ocean. Furthermore, $pCO_2$ in shelf regions is characterized by high temporal and spatial variability, making trend analyses more demanding[8–12]. The recent development of the community-driven global ocean $pCO_2$ data product SOCAT (for Surface Ocean $CO_2$ Atlas[13]) now offers a complementary approach to assess whether continental shelves show a change in the air–sea $pCO_2$ gradient ($\Delta pCO_2 = pCO_{2,air} - pCO_2$) over time. Although the data coverage remains sparse within SOCAT, it allows reconstructing the evolution in $\Delta pCO_2$ for 15 regions across the global shelves with a time span of at least a decade. We first aim to identify if the observed recent trends in $\Delta pCO_2$ support a strengthening or a weakening of the global $CO_2$ uptake by shelf regions. Then, we investigate whether important regional differences emerge from the analysis, and if any global pattern can be discerned when combining all observational evidence. In what follows, we briefly review the current state of knowledge regarding shelf $CO_2$ dynamics and then propose novel observational evidence of rates of change in the air–sea $pCO_2$ gradient from the analysis of the SOCAT database.

Syntheses in the recent decade suggest that, globally, continental shelves currently absorb atmospheric $CO_2$ at a rate of about 0.2 Pg C annually[11,12,14–18]. Despite great local variability, the data also suggest that mid- to high-latitude shelves are generally a sink of $CO_2$, while warm tropical shelves are a moderate source of $CO_2$[14,16,17]. A broad consensus regarding the current strength of the global shelf $CO_2$ sink and its large-scale spatial variability has thus recently emerged. In particular, continuous high-resolution $pCO_2$ maps for continental shelf seas derived from the interpolation of experimental data[19] clearly support this spatial trend in all oceanic basins. However, much less is known regarding decadal trends and associated variability in shelf $CO_2$ sources and sinks across the globe.

The limited $pCO_2$ time series obtained from coastal sites provide mixed evidence for the size of the decadal trends. Bates and co-authors[6] reported for the coastal stations Mundia and Iceland Sea small long-term rates of increase in $pCO_2$ (+1.3 µatm yr$^{-1}$), i.e., rate that are lower than that of the atmosphere, while they show that the stations Irminger and CARIACO have rates as high as +2.4 and +2.9 µatm yr$^{-1}$, respectively (Table 1). A shorter time series at the SEATS station in the South China Sea over the 1999–2003 period reveals an even faster increase in $pCO_2$ with a rate of +4.2 µatm yr$^{-1}$. While illustrative, such trends from a handful of locations do not allow drawing any conclusion regarding the overall change in the shelf air–sea $pCO_2$ gradient over time.

Some data-driven regional analyses have also attempted to decipher the rate of $pCO_2$ increase in continental shelf settings. Data from two large semi-enclosed shelf seas (North Sea and Baltic Sea) and from the Bering Sea suggest that continental shelves may exhibit a rapid increase in $pCO_2$[20,21] toward atmospheric values, thus lowering the air–sea $pCO_2$ gradient over time (Table 2). In contrast, another study in the North Sea[22] and reports from the warm Caribbean Sea[23] (mostly from areas deeper than the shelf depths as defined here), the coast of Japan[24], West Antarctic Peninsula[25], and the Scotia shelf[26], showed that the sea surface $pCO_2$ increase lags well behind that of the atmosphere, making the areas either an increased sink (Pacific coast of Japan, Coast West Antarctic Peninsula, and Puerto Rico) or a decreased source (Scotian shelf) for atmospheric $CO_2$. However, a recent study[27] suggests that the Japanese margin as a whole roughly tracks the atmospheric $CO_2$ increase. Overall, these regional analyses highlight that trends in $CO_2$ sources and sinks appear highly variable both within the same shelf and across different shelf systems.

Researchers have also attempted to use models to investigate the change in shelf air–water $CO_2$ exchange. Using a box model, Mackenzie and co-workers were the first to suggest that shelves may have turned from a $CO_2$ source in the preindustrial time to a sink at present and that the $CO_2$ uptake rate would increase with time[28]. Consistent with these predictions, Bauer et al.[8] and Cai[14] also provided a conceptual model and suggested an increasing global shelf $CO_2$ sink with time as a result of the atmospheric $pCO_2$ increase. Recently, an eddy-resolving global model was used to simulate the flux of anthropogenic $CO_2$ into the coastal ocean[29]. This latter model can be viewed as an open ocean model extended to the coast that lacks a few, but important processes in the nearshore environments. In particular, the global model lacks detailed sediment interactions, the handling of river fluxes, and

**Table 1 Rates of change in $pCO_2$ and corresponding rate of change in $\Delta pCO_2$ reported in the literature for coastal time series stations**

| Region | $pCO_2$ trend (µatm yr$^{-1}$) | $\Delta pCO_2$ trend (µatm yr$^{-1}$) | Period | Comment | Reference |
|---|---|---|---|---|---|
| Mundia | $1.28 \pm 0.33$ | Increase | 1988–2011 | | Bates et al.[6] |
| Hawaii (HOT) | $1.72 \pm 0.09$ | Steady | 1988–2011 | Outside of our shelf definition | Bates et al.[6] |
| | $2.0 \pm 0.5$ | Steady | 1983–2013 | | Wang et al.[27] |
| BATS | $1.69 \pm 0.11$ | Steady | 1983–2011 | Outside of our shelf definition | Bates et al.[6] |
| | $1.9 \pm 0.2$ | Steady | 1991–2011 | | Wang et al.[27] |
| ESTOC | $1.92 \pm 0.24$ | Steady | 1995–2011 | | Bates et al.[6] |
| Irminger Sea | $2.37 \pm 0.49$ | Decrease | 1983–2011 | | Bates et al.[6] |
| CARIACO | $2.95 \pm 0.43$ | Decrease | 1995–2011 | | Bates et al.[6] |
| South China Sea (SEATS) | $4.2 \pm 3.2$ | Decrease (−2.6) | 1999–2003 | Outside of our shelf definition | Tseng et al.[21] |
| Iceland Sea (year) | $1.29 \pm 0.36$ | Increase | 1995–2008 | | Bates et al.[6] |

**Table 2 Rates of change in $pCO_2$ and corresponding rate of change in $\Delta pCO_2$ in different regions reported in the literature for continental shelf waters**

| Region | $pCO_2$ trend ($\mu$atm yr$^{-1}$) | $\Delta pCO_2$ trend ($\mu$atm yr$^{-1}$) | Period | Comment | Reference |
|---|---|---|---|---|---|
| **North Sea** | | | | | |
| Summer | 7.9 | Fast decrease | 2001–2005 | Both studies only compare summertime $pCO_{2,w}$ | Thomas et al.[20]; |
| Summer | 6.5 | Fast decrease | 2001–2005 | normalized to 16 °C for different summers | Salt et al.[22] |
| Summer | 1.33 | Slow increase | 2005–2008 | | Salt et al.[22] |
| **Baltic Sea** | | Decrease | 1994–2008 | Interannual variations in $pCO_{2,w}$ minima are controlled by maximum concentration of phosphate in winter | Wesslander et al.[32] |
| **Puerto Rico** | | | | | |
| All year | 1.11 ± 0.35 | Increase (+0.74) | 2002–2009 | | Park and Wanninkhof[23] |
| Summer | 1.57 ± 0.86 | Increase (+0.47) | 2002–2009 | | Park and Wanninkhof[23] |
| Winter | 0.17 ± 1.23 | Increase (+1.88) | 2002–2009 | | Park and Wanninkhof[23] |
| **Bering Sea** | | | | Increase in $pCO_2$ attributed to abnormally high primary production | |
| Basin | 6.5 ± 1.4 | Decrease (−6.0) | 1995–2001 | | Fransson et al.[51] |
| Shelf slope | 11 ± 1.9 | Decrease (−10.0) | 1995–2001 | | Fransson et al.[51] |
| **Coast of Japan** | | | | | |
| | 1.54 ± 0.33 | Increase (+0.45) | 1994–2008 | | Ishii et al.[24] |
| | 2.1 ± 0.6 | Steady | 1992–2013 | | Wang et al.[27] |
| **Tasmanian Coast** | Increase | Not reported | 1982–2005 | Increase in $pCO_2$ is explained by sea surface temperature increase | Borges et al.[48] |
| **European Margins** | 1.9 ± 0.7 | Steady | 1989–2014 | | Wang et al.[27] |
| **Antarctic Peninsula** | | | | | |
| Summer | 1.45 ± 2.97 | Increase (+0.45) | 1999–2013 | | Hauri et al.[25] |
| Fall | 1.90 ± 0.95 | Steady | 1999–2013 | | Hauri et al.[25] |
| Winter | 0.43 ± 0.77 | Increase (+1.47) | 1999–2013 | | Hauri et al.[25] |
| Spring | 1.22 ± 2.72 | Increase (+0.68) | 1999–2013 | | Hauri et al.[25] |
| **Scotia Shelf** | Not reported | Increase (+2.3) | 1999–2008 | The increase in $\Delta pCO_2$ is attributed to an increase of 1.3 °C in sea surface temperature | Shadwick et al.[26] |

shallow calcification processes, which were captured in the spatially and temporally crude box model, however[28,30]. Nonetheless, both approaches consistently show that the shelf water $CO_2$ uptake increases with increasing atmospheric $CO_2$ levels. However, no consensus emerges as to whether past and future decadal changes in shelf $\Delta pCO_2$ and, thus, $CO_2$ absorption per unit area, will increase at a faster or slower rate compared with the global open ocean.

Two main mechanisms have been proposed to explain the evolution of the continental shelf $CO_2$ sink. The first mechanism relies on the efficiency of the physical pump and more specifically on the different timescales for the air–water and shelf-open ocean exchanges of $CO_2$[8,14]. In the situations where the $CO_2$ exchange rate across the shelf is faster than that with the atmosphere, the $pCO_2$ increase in waters on the shelves may be slower than the atmospheric $pCO_2$ increase, even if we assume that no change in biology and physics occurs over time[8,14]. In these margins, the accumulation of anthropogenic $CO_2$ in shallow waters would be limited and would help maintain a significant air–water $pCO_2$ gradient favouring an efficient uptake of anthropogenic $CO_2$. In contrast, if the cross-shelf export is unable to keep up with the increasing air–sea flux of anthropogenic $CO_2$, $CO_2$ in shelf waters may accumulate and the $pCO_2$ increase would follow the atmosphere due to this bottleneck in offshore transport[29].

The second mechanism relies on the stimulation of the biological pump. Many continental shelves are seriously influenced by anthropogenic nutrient inputs and have higher biological production today than what they had in preindustrial time[30,31]. Thus, net ecosystem metabolism (NEM) on the shelves could have progressively shifted from net heterotrophy to net autotrophy and the change could have been sufficiently large to reverse the air–sea $CO_2$ flux from a source during preindustrial times to a sink under present-day conditions. Net ecosystem calcification (NEC) also plays a significant role in the air–water $CO_2$ exchange on the shelves, but the contribution of the carbonate pump to changes in air–water exchange fluxes over the historical period are likely not as large as the biological pump[28].

Our aim here is to present the first observation-based analysis of decadal trends in global shelf $pCO_2$. The results presented in our regional and global analysis are primarily derived from wintertime data when photosynthetic activity is generally the weakest, and when coastal ocean waters have the most intensive exchange with the open ocean, and consequently the strongest impact on the global ocean $CO_2$ accumulation[17,31]. This results in trends that tend to be clearer. We check on these wintertime analyses also with results from an analysis using deseasonalized data for all seasons, confirming that our choice for wintertime only does not result in artefacts. However, this does not suggest that winter contributes more than other seasons to the overall annual trend.

## Results

**Regional trends in $\Delta pCO_2$.** Our analysis employing a narrow definition of the continental shelf corresponding to the 200 m isobaths and winter-only data provides decadal trends in $\Delta pCO_2$, i.e., $d\Delta pCO_2/dt$ values, for 825 cells with an average length of our time series of 18 years. Six hundred five of these 825 cells belong to 6 large regions each comprising at least 50 cells (Table 2). Another 190 cells belong to 9 smaller regions each comprising 10 cells or more. Together, these 15 regions account for 96 and 80%

**Table 3 Rates of change in $\Delta p\text{CO}_2$ in different regions calculated as the average of the rates derived for each cell of the region**

| Region | Narrow shelf | | | Wide shelf | | |
|---|---|---|---|---|---|---|
| | $d(\Delta p\text{CO}_2)/dt$ | $\sigma$ | $n$ | $d(\Delta p\text{CO}_2)/dt$ | $\sigma$ | $n$ |
| Large regions (>50 cells according to narrow shelf definition) | | | | | | |
| North Sea | 1.86 | 1.55 | 169 | 1.81 | 1.37 | 186 |
| Baltic Sea | 2.93 | 2.38 | 114 | 2.93 | 2.38 | 114 |
| Labrador Sea | 0.68 | 0.61 | 104 | 0.71 | 0.67 | 115 |
| English Channel | 0.0 | 0.43 | 86 | −0.03 | 0.39 | 89 |
| Mid-Atlantic Bight | 1.93 | 3.11 | 76 | 1.92 | 3.19 | 78 |
| Coast of Japan | 0.77 | 0.69 | 56 | 0.22 | 0.7 | 175 |
| Small regions (>20 cells according to narrow shelf definition) | | | | | | |
| Cascadian shelf | 0.83 | 1.72 | 27 | 0.97 | 1.23 | 49 |
| Patagonia | −0.21 | 0.38 | 27 | −0.11 | 0.35 | 33 |
| Irminger Sea | 0.56 | 0.23 | 26 | 0.47 | 0.35 | 35 |
| Bering Sea | −1.11 | 0.74 | 24 | −1.44 | 0.94 | 42 |
| Antarctic Peninsula | 2.28 | 1.24 | 22 | 1.57 | 0.95 | 50 |
| South Greenland | 1.95 | 1.22 | 20 | 1.73 | 0.8 | 29 |
| South Atlantic Bight | 0.51 | 0.74 | 18 | 0.7 | 0.7 | 26 |
| Tasmania | 0.11 | 0.12 | 16 | 0.15 | 0.17 | 25 |
| Barents Sea | 0.38 | 0.52 | 10 | 0.31 | 0.41 | 42 |

The standard deviation ($\sigma$) and the number of cells available ($n$) using our narrow and wide definitions of the continental shelf are also reported

of the cells contained in our narrow and wide definitions of the shelf, respectively (see "Methods" section for details). Most regions display variable, but relatively consistent values of $d\Delta p\text{CO}_2/dt$. Only the Baltic Sea and the Mid-Atlantic Bight show a significant but continuous spatial gradient in the trend within their respective domains. With the exception of the Labrador Sea, regional analyses of the air–sea $\text{CO}_2$ exchange have been published for each of the areas presented here, but estimates of $d\Delta p\text{CO}_2/dt$ have only been reported for 9 out of the 15 regions (Table 3).

The highest positive $d\Delta p\text{CO}_2/dt$, with an average rate of change of +2.9 ± 2.4 µatm yr$^{-1}$, occurs in the Baltic Sea (Fig. 1a). This rate of increase in $\Delta p\text{CO}_2$ is higher than the atmospheric $p\text{CO}_2$ increase rate and, therefore, surface water $p\text{CO}_2$ actually decreases over time, most likely as a result of increased anthropogenic nutrient inputs and resultant increases in coastal productivity in this semi-isolated inland sea, which affect the biogeochemistry of the Baltic Sea all year long[32]. Our results are consistent with a recent study that the Baltic Sea is a decreasing source of $\text{CO}_2$ to the atmosphere[32]. The North Sea (Fig. 1b), the Mid-Atlantic Bight (Fig. 1c), Southern Greenland, and Antarctic Peninsula (Fig. 1d) have $d\Delta p\text{CO}_2/dt$ values averaging close to +2 µatm yr$^{-1}$ with $\sigma$ of 1.2–1.5 µatm yr$^{-1}$, except for the Mid-Atlantic Bight, which has more spatial heterogeneity ($\sigma = 3.1$ µatm yr$^{-1}$). Therefore, their water $p\text{CO}_2$ values do not increase with time or increase at a rate substantially lower than that of $p\text{CO}_{2,\text{air}}$ (Table 3). The results for the North Sea contrast sharply with an earlier report based on two sets of summertime data (2005 vs. 2001), which suggested that the North Sea $p\text{CO}_2$ increased at a rate five times faster than the atmosphere[20]. A more recent study, however, also comparing summertime $p\text{CO}_2$ between the years 2001, 2005, and 2008 revealed a large increase of 26 µatm between 2001 and 2005, but only a moderate increase of 4 µatm between 2005 and 2008[22]. These disparate results support the idea that summertime $p\text{CO}_2$ is more affected by the short-term imbalance of biological production and respiration, whereas wintertime reflects better the long-term trend in air–sea exchange due to reduced biological activities. The increase in the $\text{CO}_2$ uptake by Greenland coastal waters reported here is in agreement with Yasunaka et al.[33]. Along the East coast of the United States, the Mid-Atlantic Bight is a typical western boundary current margin with intense exchange of water between the shelf and the deep

ocean at a frequency of about once every 3 months and is also influenced by anthropogenic nutrient inputs and eutrophication[31,34]. A previous study suggested that the annual thermal cycle combined with high winds during wintertime dominates annual $\text{CO}_2$ uptake in this region[35]. $d\Delta p\text{CO}_2/dt$ ranging from values >+5 µatm yr$^{-1}$ in the north of the region to <−2 µatm yr$^{-1}$ in the south could be a consequence of very different hydrodynamic characteristics along this coastal setting. The southern part of the region is under the influence of coastal currents and large estuaries (e.g., Chesapeake Bay) that effectively filter terrestrial organic carbon inputs[36,37] while the northern part is characterized by significantly colder water from the Labrador Sea all year long[38]. The Antarctic margins, such as those along the Antarctic Peninsula, are dominated by an intense exchange with deep water masses and it appears that the rate of $\text{CO}_2$ uptake is driven mainly by the strength of the upwelling and low-surface temperature[39]. The high $d\Delta p\text{CO}_2/dt$ along the West Antarctic Peninsula is consistent with another study[25], which also suggested that winter is the season for which the rate of increase in $\Delta p\text{CO}_2$ is the fastest. Additionally, a general strengthening of the Southern Ocean $\text{CO}_2$ sink has been recognized over the past decade[39].

A second group of regions includes the shelves of Irminger Sea and the Labrador Sea, the Coast of Japan (Fig. 1e), the Cascadian shelf (Fig. 1f), and the South Atlantic Bight. These shelf regions have $d\Delta p\text{CO}_2/dt$ values ranging between +0.5 µatm yr$^{-1}$ and +1.0 µatm yr$^{-1}$. This range implies that their water $p\text{CO}_2$ is increasing, but at a rate that is moderately slower than that of $p\text{CO}_{2,\text{air}}$, implying a strengthening sink, or a weakening source. The South Atlantic Bight is a moderate sink of $\text{CO}_2$ for the atmosphere[40] because of its water residence time of a few months and rapid cross-shelf exchange with the open ocean in the winter[41]. Our estimate of +0.8 µatm yr$^{-1}$ for the Pacific coast of Japan is also consistent with a survey[24] that reports a slower increase of water $p\text{CO}_2$ (+1.5 ± 0.3 µatm yr$^{-1}$, Table 2) than that of $p\text{CO}_{2,\text{air}}$ (+2.0 ± 0.1 µatm yr$^{-1}$) over the period of 1994–2008. Perhaps a bit counterintuitive, however, is the low-positive $d\Delta p\text{CO}_2/dt$ (+0.8 ± 1.7 µatm yr$^{-1}$) along the Eastern Boundary current margins known for their strong upwelling off the U.S. West Coast (the California and Cascadian shelves). However, here upwelling source waters are not from the deeper Antarctic water as that in the Atlantic Ocean; rather, they are North Pacific surface water

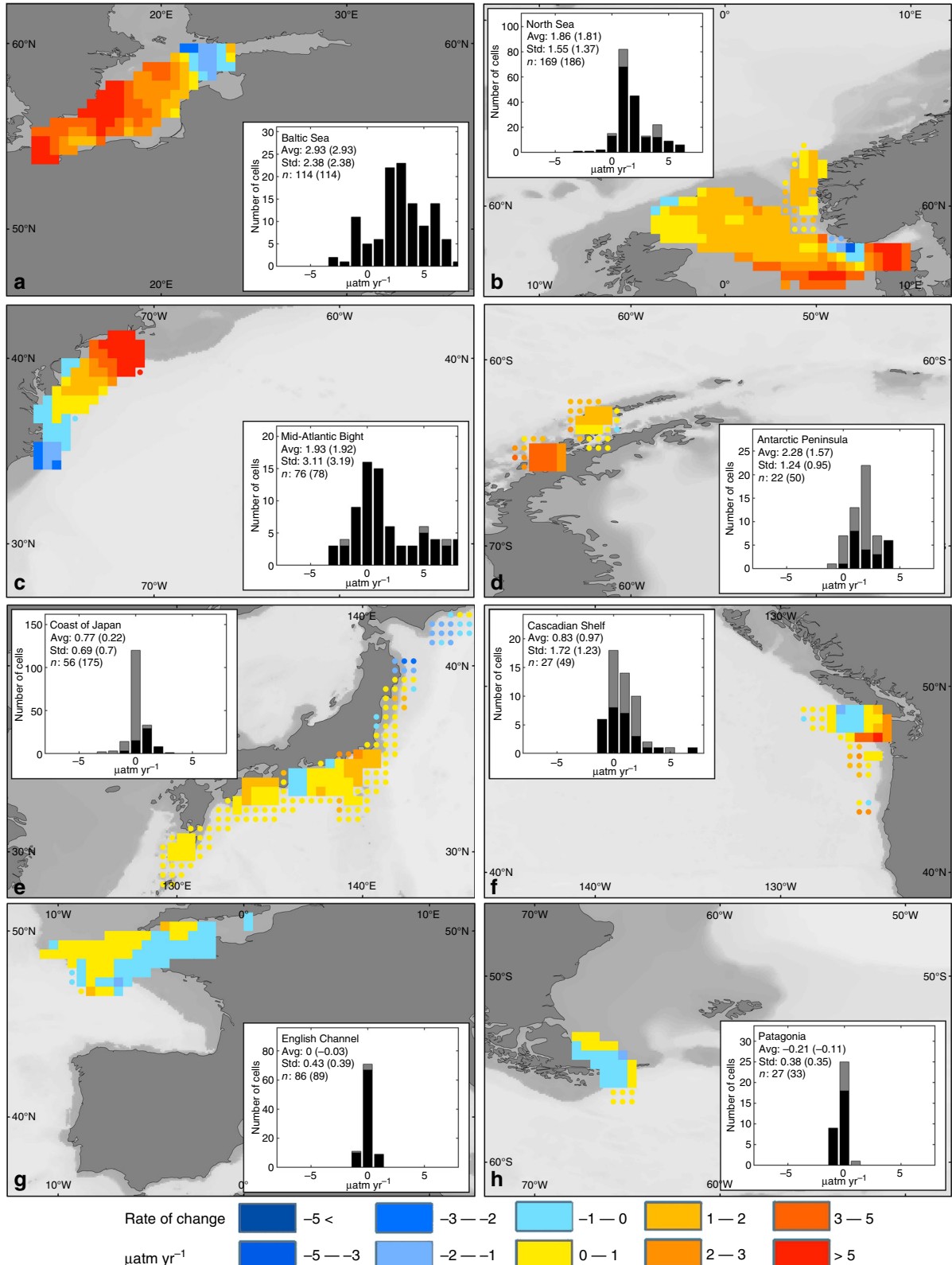

**Fig. 1** Rate of increase in winter air–sea $p$CO$_2$ gradient for selected best surveyed continental shelf regions. **a** Baltic Sea, **b** North Sea, **c** Mid-Atlantic Bight, **d** Antarctic Peninsula, **e** Coast of Japan, **f** Cascadian shelf, **g** English Channel, and **h** Patagonia. Each point represents a 0.5°x0.5° cell. Cells belonging to the narrow and wide definition of the shelf are displayed as squares and diamonds, respectively. The inserted histogram provides the distribution of the average d($\Delta p$CO$_2$)/d$t$ value, standard deviation, and the number of cells using the narrow (black lines) and wide (gray lines and numbers in brackets) definitions of the shelf

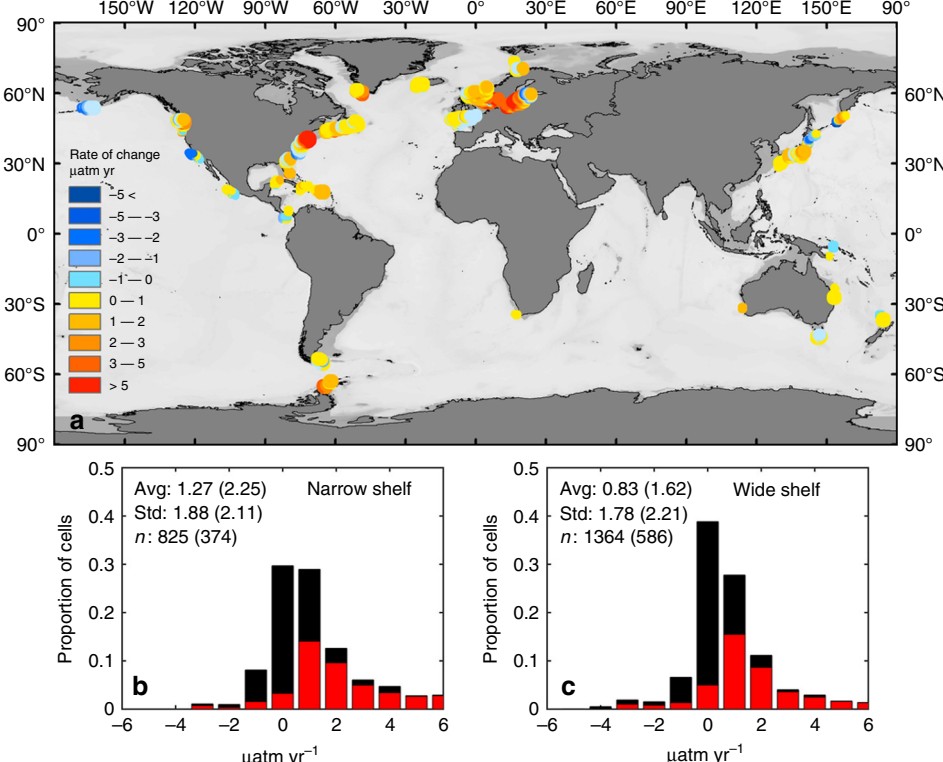

**Fig. 2** Location of 0.5° cells for which the decadal trend in winter $\Delta pCO_2$ is calculated (**a**). Large dots correspond to cells shallower than 200 m and small dots correspond to cells located within 100 km from the coast or depth less than 500 m. The distribution of d($\Delta pCO_2$)/dt for both our narrow (**b**) and wide (**c**) definitions of the continental shelf are displayed as histograms. The black bars report the distribution of all cells while the red bars report the distribution of cells for which the trend was deemed statistically significant using an $F$-test with $p < 0.05$. Here, $\Delta pCO_2 = pCO_{2,air} - pCO_2$. Thus, positive values in d($\Delta pCO_2$)/dt indicate slower increase in water $pCO_2$ than $pCO_{2,air}$

subducted only decades earlier, which thus carries with it the anthropogenic $CO_2$ signal[42–44]. The enhanced upwelling strength in recent years may also have contributed to an increase in sea surface $pCO_2$[45]. The western shelves of North America especially the Cascadian shelves, however, are known for their strong spatial heterogeneity, suggesting that multiple processes drive their biogeochemical behavior[46].

The third group of regions, which includes the English Channel (Fig. 1g), the Barents Sea, and the Tasmanian shelf, show a minimal or no increase in d$\Delta pCO_2$/dt, meaning that their water $pCO_2$ more or less tracks the $pCO_{2,air}$ increase (Table 3). The interannual dynamics of $pCO_2$ in the English Channel is largely influenced by North Atlantic waters and thus partly constrained by the North Atlantic Oscillation[47]. While no long-term d$\Delta pCO_2$/dt has been estimated for the English Channel itself, the increase of $+1.7\,\mu atm\,yr^{-1}$ for $pCO_2$ calculated for adjacent Atlantic water[1] is consistent with an increase following that of the atmosphere. In both the Barents Sea[33] and Tasmanian shelf[48], signs of a strengthening of the coastal $CO_2$ sink have been reported and were partly attributed to decreases in sea surface temperature. While marginal ($+0.1\,\mu atm\,yr^{-1}$), the d$\Delta pCO_2$/dt revealed by our calculations suggests an increase in the strength of $CO_2$ sink in the Tasmanian shelf. Finally, the Patagonian continental shelf ($-0.2 \pm 0.4\,\mu atm\,yr^{-1}$; Fig. 1h) and Bering Sea ($-1.1 \pm 0.7\,\mu atm\,yr^{-1}$) are the only regions displaying a negative d$\Delta pCO_2$/dt (meaning faster increase in water $pCO_2$ than air $pCO_2$). Thus, although they are still intense $CO_2$ sinks[17,49,50], these shelf systems have recently experienced a weakening in their capacity to absorb atmospheric $CO_2$. While no long-term trend was reported for Patagonian shelf, it has been suggested that the $CO_2$ uptake in the Bering Sea could be decreasing at a fast pace

although these observations were based on a relatively short time series (1995–2001)[51].

Overall, 13 of the 15 regions have positive d$\Delta pCO_2$/dt values and 10 reveal values equal or greater than $+0.5\,\mu atm\,yr^{-1}$ (Table 3). Although these areas only account for a small fraction of the global coastal ocean, they show a consistent trend suggesting that winter sea surface $pCO_2$ increases significantly slower than $pCO_{2,air}$. Furthermore, in most regions, the variability around this trend is relatively limited. For instance, in 9 out of 15 regions, the standard deviation is less than $1\,\mu atm\,yr^{-1}$. However, it remains difficult to identify the mechanisms responsible for these observed patterns in d$\Delta pCO_2$/dt considering the diversity of morphological and hydrodynamical settings of the shelf regions covered by our analysis.

**Global shelf CO₂ sink.** Globally, our analysis of the 825 temporal trends in $\Delta pCO_2$ using winter-only data covers a shelf surface area of $1.4 \times 10^6\,km^2$, which represents ~6% of the global continental shelves. This includes data from the more isolated cells that were not considered in the previous section. While the coverage is relatively small, heterogeneous and somewhat skewed toward temperate latitudes in the northern hemisphere, it nevertheless covers most of the range of $pCO_2$ and SST encountered in continental shelf waters (Supplementary Figures 1 and 2). Exceptions are the low latitudes, which are poorly represented in our data set. While we need to recognize this limitation, the broad coverage in terms of environmental conditions permits us to assemble all estimated trends and assume that they represent a sufficiently unbiased sample of the shelf trends across the globe.

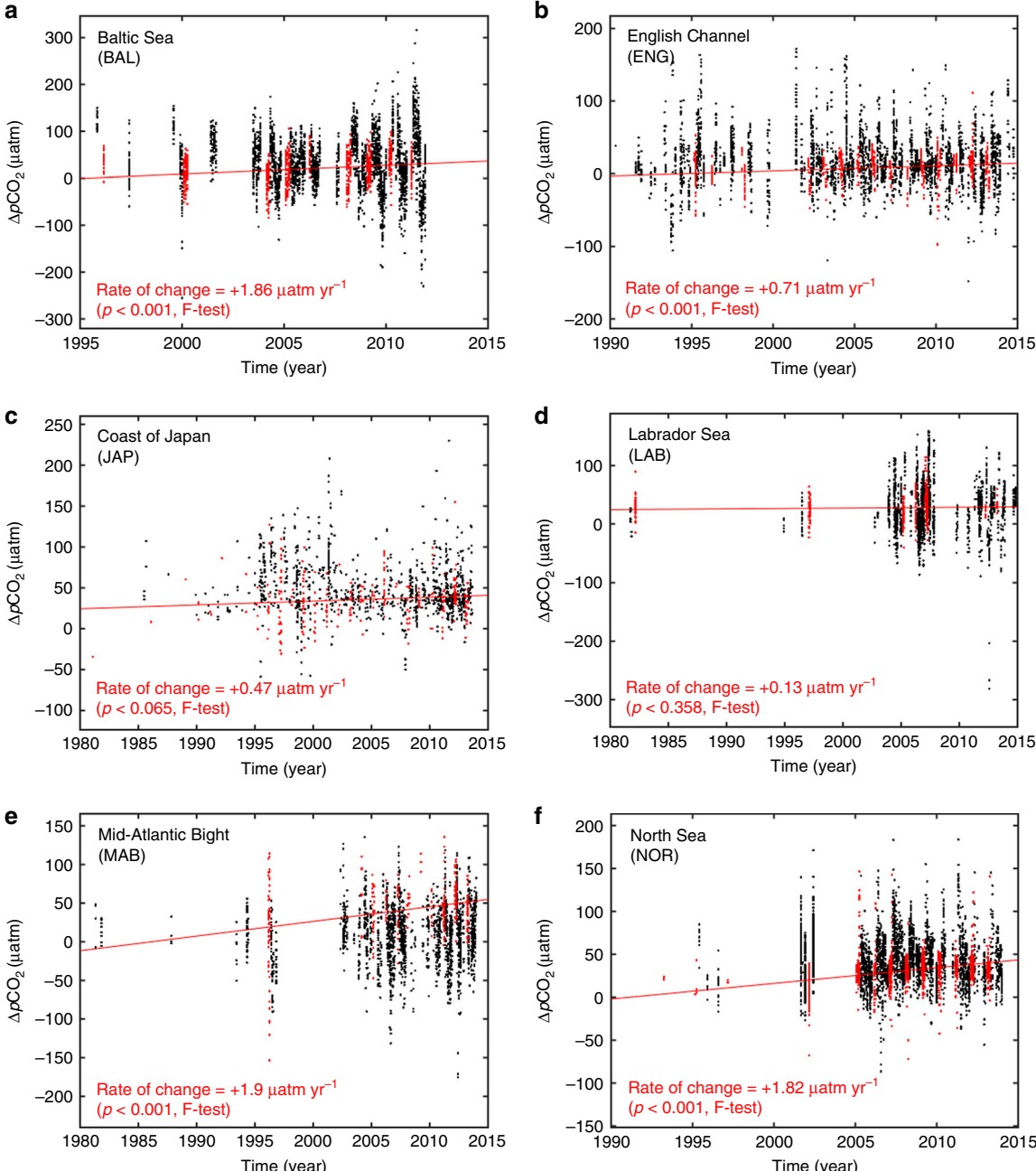

**Fig. 3** $\Delta p\mathrm{CO_2}$ as a function of time for all cells comprised in the six regions with best data coverage. **a** Baltic Sea, **b** English Channel, **c** Coast of Japan, **d** Labrador Sea, **e** Mid-Atlantic Bight, and **f** North Sea. Red dots correspond to winter data only, black dots to data for all seasons

The bulk of our winter data consistently show trends that are dominantly in accordance with an increase in $\Delta p\mathrm{CO_2}$ over time. Our narrow definition of the shelf yields a global average $\mathrm{d}\Delta p\mathrm{CO_2}/\mathrm{d}t$ of $+1.3 \pm 1.9\ \mu\mathrm{atm\ yr^{-1}}$, while the wide definition for geographical extent leads to a smaller average value of $+0.8 \pm 1.8$ $\mu\mathrm{atm\ yr^{-1}}$ (Fig. 2). Thus, our global-scale analysis of winter $p\mathrm{CO_2}$ data reveals that trends are more likely positive than not and support the idea that air–sea $p\mathrm{CO_2}$ gradients may have been increasing with time making continental shelves overall an increasing $\mathrm{CO_2}$ sink for the atmosphere. Nevertheless, as shown by the substantial standard deviations, large differences in $\mathrm{d}\Delta p\mathrm{CO_2}/\mathrm{d}t$ can be observed across continental shelves. Within the 200 m water depth boundary, 653 cells (out of 825) display a positive $\mathrm{d}\Delta p\mathrm{CO_2}/\mathrm{d}t$, 76% of which are greater than $+0.5\ \mu\mathrm{atm\ yr^{-1}}$ (i.e., 495 out of 653; Supplementary Table 1). For 66% of the

latter cells (325 out of 495), the slope of the regression is considered statistically significant using an $F$-test with $p < 0.05$ and 71% with $p < 0.1$ (Fig. 2). On the other hand, for the 172 cells (out of 825) that display negative $\mathrm{d}\Delta p\mathrm{CO_2}/\mathrm{d}t$ values, only 49% are more negative than $-0.5\ \mu\mathrm{atm\ yr^{-1}}$ (84 out of 172). The trend is still observed when the boundary is relaxed to 500 m or 100 km from the coast. One thousand and sixty-six cells (out of 1364) display a positive $\mathrm{d}\Delta p\mathrm{CO_2}/\mathrm{d}t$, 64% of which is greater than $+0.5$ $\mu\mathrm{atm\ yr^{-1}}$ (i.e., 682 out of 1066) and only 149 out of the 298 non-positive cells having a negative $\mathrm{d}\Delta p\mathrm{CO_2}/\mathrm{d}t$ are then characterized by rates more negative than $-0.5\ \mu\mathrm{atm\ yr^{-1}}$ (Fig. 2). The use of the broader definition of the continental shelves not only decreases the average $\mathrm{d}\Delta p\mathrm{CO_2}/\mathrm{d}t$, but also increases the proportion of cells with $\mathrm{d}\Delta p\mathrm{CO_2}/\mathrm{d}t$ between $-0.5$ and $+0.5$ $\mu\mathrm{atm\ yr^{-1}}$ (39% vs. 30%). Note that applying our method to all

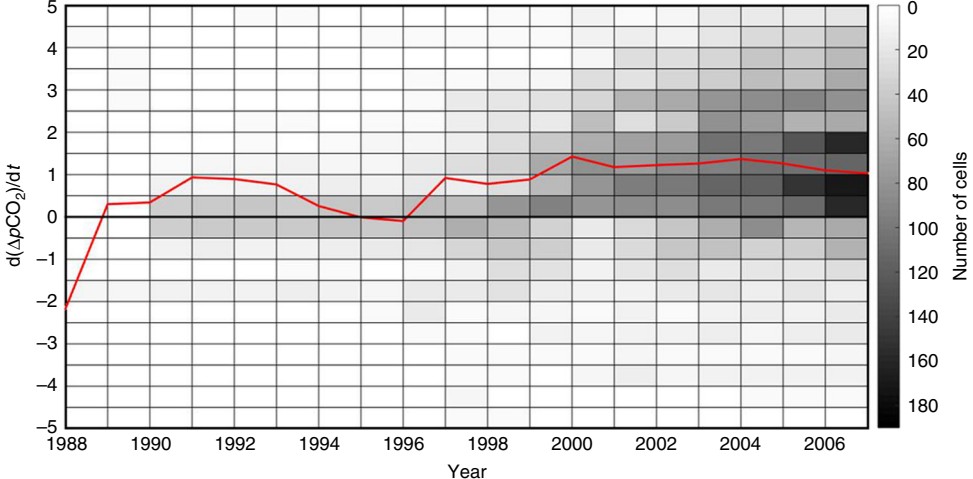

**Fig. 4** Evolution of winter air–sea $pCO_2$ gradient for the global shelves over the 1988–2007 period. The gray scale shows, every year, the number of cells for different ranges of $d(\Delta pCO_2)/dt$. The red line provides the evolution of the average $d(\Delta pCO_2)/dt$ for all cells with data over time

open ocean waters deeper than 500 m or further than 100 km from the coast yields a much smaller average $d\Delta pCO_2/dt$ of +0.2 ± 1.1 µatm yr$^{-1}$, which is close to the open ocean observation[1–6,52], further supporting the validity of our method.

The results from the analysis using data across the entire year generally confirm the results from the wintertime-only data as exemplified by the change in $\Delta pCO_2$ with time for all cells pertaining to the six largest regions used in the regional analysis (Fig. 3). For all regions, the range of $\Delta pCO_2$ values observed in winter (red) is largely less than that based on the entire year (black). Globally, calculations performed using deseasonalized data from the entire year allow deriving trends for 3721 cells (Supplementary Table 1). Although much noisier, the overall $d\Delta pCO_2/dt$ values using all seasonal data reveal qualitatively similar trends to those observed with data from the winter months only (Supplementary Table 1). The overall proportion of cells displaying statistically significant trends is much lower when all seasonal data are used (22%) than when winter data are retained (45%). Nevertheless, nearly three times more cells display significant trends for which $d(\Delta pCO_2)/dt > +0.5$ µatm yr$^{-1}$ (574) than for which $d(\Delta pCO_2)/dt < -0.5$ µatm yr$^{-1}$ (201), a result in broad agreement with our analysis based on winter data only. Therefore, the analysis of deseasonalized data for all seasons also point toward a tendency for an enhanced shelf uptake of atmospheric $CO_2$.

The rate of change in the air–sea $CO_2$ gradient also has varied over time. Figure 4 presents the evolution from 1988 to 2007 of winter $d\Delta pCO_2/dt$ calculated for each year over a 15-year time period. Because the bulk of the data available in SOCAT are relatively recent, it is difficult to reconstruct trends earlier than the 1990s. The distribution of rates around the mean value is shown by the gray scale in Fig. 4 and the widening of the distribution can be observed as the number of data points increase but, for any given year, the bulk of the $d\Delta pCO_2/dt$ distribution remains constrained within the $-0.5$ to $+2.0$ µatm yr$^{-1}$ range. While uncertainties are high in such an analysis, in particular because the trends from the investigated regions (about 6% of the total shelf area) might not hold for all the others, our results suggest that in addition to the dominance of positive $d\Delta pCO_2/dt$, there is a good probability that the rate of change of the air–sea $CO_2$ gradient has also increased over the past 15 years. Indeed, the average $d\Delta pCO_2/dt$ appears to remain below 1 µatm yr$^{-1}$ before 1997 but above it since then.

**Implications for the global carbon budget**. Applying the mean rate of change in the winter air–sea $CO_2$ gradient identified in Fig. 4 to a globally averaged winter $\Delta pCO_2$ of +28 µatm for the continental shelf seas in the reference year 2000[17], leads to an increase in water $pCO_2$ that has consistently lagged behind the increase in atmospheric $CO_2$. As noted in the previous section, $d\Delta pCO_2/dt$ also has increased in recent years (+1.2 µatm yr$^{-1}$). The first records of shelf $pCO_2$ date back from the early 1980s and it is thus impossible to reconstruct the earlier evolution of the air–sea exchange from direct observations. Although highly speculative, it is nevertheless interesting to extend the rate of change in the 1980s and early 1990s (+0.6 µatm yr$^{-1}$) estimated here to the earlier decades. This approach allows comparing trends derived from observations alone with earlier modeling work and will stimulate further exploration of coastal $CO_2$ trend[28,30]. Our calculations suggest that the magnitude of the average winter $\Delta pCO_2$ increased by 69% over the 1988–2007 period. By extrapolating this trend to earlier times, it can be speculated that the continental shelves might indeed have turned from a purported preindustrial source of $CO_2$ into a sink for atmospheric $CO_2$ as early as in the 1950s, at least during wintertime (Supplementary Figure 3). The occurrence of such a switch from source to sink in the mid-twentieth century would be consistent with previous model results[28], although our data-based estimate may indicate that the switchover time could have occurred earlier than previously thought. Note that our assessment excludes the high $pCO_2$ estuarine and very nearshore (proximal) zone that is believed to be a significant source of $CO_2$ for the atmosphere at present[12,16,53,54]. A recent model hindcasts that the uptake rate of anthropogenic $CO_2$ by continental shelves has increased rapidly since the 1950s[29]. Although this flux is much less than that modeled for the open ocean uptake[29], it would imply an increase in the total $CO_2$ uptake flux (natural plus anthropogenic) and an increase in the air–sea $pCO_2$ gradient in the coastal ocean, assuming that the natural $CO_2$ flux in their model does not change. Thus, the conjecture derived from our first global observation-based work is consistent with their model prediction. Nevertheless, this work also highlights the fact that the rate of increase of the air–sea $pCO_2$ gradient in continental shelf waters and its importance in global ocean $CO_2$ uptake is still poorly understood and deserve further study. In addition, if both observational evidence and model results support the idea that shelves are an increasing sink for the atmospheric $CO_2$, we are far from quantitatively understanding the roles of the physical pump

and biological pumps (NEM and NEC) in explaining this enhanced $CO_2$ absorption and associated high variability globally. We suggest, however, that a faster exchange of shelf $CO_2$ with the ocean interior and increased biological production due to anthropogenic nutrient inputs may have slowed down the rate of increase of surface ocean $pCO_2$ in many shelf regions.

In principle, the slower $pCO_2$ increase in shelf waters could increase the gradient and thus the uptake of atmospheric $CO_2$ in the decades to come, although high spatial variability in air–sea fluxes is to be expected across shelf regions. The shift of shelf waters from releasing to absorbing $CO_2$ between preindustrial time and the present day, as well as the possibility of shelves becoming a more important sink in the future, is a significant temporally changing term in the global carbon cycle and pathway of exchange for atmospheric $CO_2$. It should thus be closely evaluated by further data collection and analysis and be considered in future global carbon cycle models and flux assessments[6,31,35,39,52].

## Methods

**Definition of the study area**. For this work, we defined the continental shelf as all marine waters shallower than the 200 m isobath. This depth is commonly used in the literature as the depth at which the shelf breaks[12,16,17,31,52]. However, we also report results for a less restrictive definition, where the shelf limit is set at 500 m water depth or within 100 km from the shore (Supplementary Figure 4). This allows inclusion of the wide and deep shelves at high latitudes and of coastal processes that take place in deeper waters in regions where the shelf break is very close to the shore. With both definitions, coastal waters shallower than 20 m as well as internal waters, such as estuaries, fjords, lagoons, or tidal marshes, are excluded from this analysis. The $pCO_2$ data selection was performed using water depths extracted from ETOPO2 and the distance to the coast was provided by SOCAT3. Our narrow and wide definitions of the continental shelf correspond to global surface areas of $22 \times 10^6$ $km^2$ and $45 \times 10^6$ $km^2$, respectively. These two values can be considered as lower and upper bounds as they comprise all reported surface areas obtained with other definitions of the continental shelf[52].

**Data processing**. Recently, numerous continental shelf $pCO_2$ observational data have been quality controlled and included into the SOCAT database[13]. Version 3 released in 2015 comprises more than $14 \times 10^6$ measurements for the entire ocean, of which $3.4 \times 10^6$ and $5.2 \times 10^6$ are located within our narrow and wide shelf definitions, respectively. This unprecedented data coverage offers the opportunity to assess whether global shelf waters show a change in the direction and magnitude of the air–sea $pCO_2$ gradient ($\Delta pCO_2 = pCO_{2,air} - pCO_2$) over time (d$\Delta pCO_2$/d$t$).

The coastal zone was divided into regular $0.5 \times 0.5$ degree cells and all the SOCAT measurements were allocated to a given cell according to their latitudes and longitudes. SOCAT fugacity data ($fCO_2$) were converted into $CO_2$ partial pressure ($pCO_2$) in water using the following equation[55].

$$(pCO_2) = fCO_2 \left(1.00436 - 4.66910^{-5} \, SST\right), \quad (1)$$

where SST is the sea surface temperature in degrees Celsius. For each month, an average $\Delta pCO_2$ was calculated within each cell. The winter data (defined as January, February, and March in the Northern Hemisphere and July, August, and September in the Southern Hemisphere) are not modified prior to calculating the linear regressions. The data from all seasons, however, are deseasonalized using monthly $pCO_2$ climatological maps for continental shelves generated by artificial neuronal network interpolations[19]. This monthly $pCO_2$ climatology allowed establishing an average seasonal $pCO_2$ cycle for each grid cell. This signal was then removed from the raw data to perform a deseasonalization prior to calculate the linear regressions.

We found no significant trend in SST in the majority of the cells (i.e., the absolute rate of change in temperature only exceeds 0.1 °C yr$^{-1}$ in less than 15% of the cells) and the average temperature change among all 825 cells used for the winter analysis using the narrow definition of the shelf is –0.0021 °C yr$^{-1}$. In addition, warming should lead to a higher water $pCO_2$ and, thus, should reduce d$\Delta pCO_2$/d$t$.

For each data point, an atmospheric $pCO_2$ was also calculated using:

$$(pCO_2)_{air} = XCO_2(P_{baro} - P_{sw}), \quad (2)$$

where $P_{baro}$ is the barometric pressure at sea surface and $P_{sw}$ is the water pressure at the temperature and salinity of the water within the mixed layer. $XCO_2$ is the weekly mean $CO_2$ concentration for dry air extracted from the GLOBALVIEW-$CO_2$ database[56]. $P_{sw}$ was calculated assuming 100% humidity using sea surface temperature and salinity and $P_{baro}$ is the monthly mean barometric pressure at the

sea surface from the NCEP/NCAR Reanalysis database[57]. All the data used were taken from the SOCAT files.

A moving spatial window of 1.5 degrees of width (i.e., three cells) was used to increase the data pool and minimize the effect of anomalous single measurements on our calculations. Effectively, this means that, for a given grid cell, all data located in the surrounding eight cells are also taken into account in the regression calculation[58]. Then, within each 0.5 degree cell, the slope of the linear regression of $\Delta pCO_2$ vs. time using the bi-square method was calculated to evaluate d$\Delta pCO_2$/d$t$. Additionally, to reduce the influence of interannual variations in $\Delta pCO_2$, we limited the analysis to cells for which we could extract at least ten data points from five or more different years spanning a period of at least 10 years between the first and last measurement. These operations are performed in similar fashion for winter-only and all-year deseasonalized data. Although the minimum period is short compared to any decadal variability that might be present in the trend, it was chosen to keep a sufficiently large pool of cells in the statistical analysis. Linear regressions were then calculated using the bi-square method. The slope of the linear regression provides the rate of change in $\Delta pCO_2$, which is defined as

$$d(\Delta pCO_2)/dt = (\Delta pCO_2(t_2) - \Delta pCO_2(t_1))/(t_2 - t_1) \quad (3)$$

where $t_2-t_1$ defines the period for which winter data are available for a given cell.

Finally, the cells for which d$\Delta pCO_2$/d$t$ could be calculated are clustered into relatively broad regions consisting of groups larger than 50 connected cells and smaller regions consisting of groups larger than 10 connected cells following our narrow definition of the shelf (Supplementary Figure 5). These groups are used as a basis for the regional analysis and allows, for the first time, to analyze consistently temporal trends in winter $\Delta pCO_2$ over the global continental shelves in a similar fashion.

**Statistics**. Prior to calculating the linear regressions, several statistical tests were performed for each cell. First, the normality of the distribution of the residuals was evaluated using a Kolmogorov and Smirnov test and >95% of the time series do not exhibit significant deviations from a normal distribution (Supplementary Figure 6a). Then, the existence of autocorrelation within the time series has been diagnosed using a Durbin–Watson's test. Results yield values clustered around 2, revealing no significant autocorrelation (Supplementary Figure 6b). The consistency of the variance of the residuals over time was assessed using White's test and significant heteroscedasticity was detected in about half of the times series. As a consequence, the linear regressions were calculated using the bi-square method rather than the simple least square method, which is less robust when the residual is not consistent throughout the entire data set.

These statistical tests were also used to determine the minimum required length of the time series (i.e., data spanning at least 10 years, more than 10 individual data, at least 5 different years with data). Using this set of criteria, the average length of our time series is 18 years. Finally, to assess the statistical significance of the regressions, an F-test was performed in each cell (Supplementary Table 1).

**Temporal evolution of $\Delta pCO_2$**. To investigate how the rate of change in air–sea $CO_2$ gradient, d$(\Delta pCO_2)$/d$t$, has varied globally over time, the entire SOCAT data set was used to produce 20 subsets, each covering a period of 15 years (from 1980 to 1994, then 1981 to 1995 and so on until 2001 to 2015) and used to calculate the rate of change for the central year of each period. For instance, d$(\Delta pCO_2)$/d$t$ for year 1988 is calculated using data ranging from 1981 to 1995. This method provides estimates for years 1988 to 2006. For each period, d$(\Delta pCO_2)$/d$t$ were calculated for each cell following the procedure described above. The average rate of change in $\Delta pCO_2$ for a given period was then calculated as the average of all the rates calculated for each cell for which observations were available (Fig. 4).

A global estimate for the coastal ocean carbon sink for winter of 0.26 Pg C yr$^{-1}$[17] was used over a surface area of $30 \times 10^6$ $km^2$ in conjunction with the gas transfer velocity of 9.0 cm h$^{-1}$ to estimate an average global $\Delta pCO_2$ of 28 µatm for the continental shelf seas in the reference year 2000. Year 2000 was selected as the estimate of Laruelle et al.[17] represents an average over the 1990–2011 period. From this reference value, the $\Delta pCO_2$ in previous and following years (period 1988–2006) was calculated by adding or subtracting the average d$(\Delta pCO_2)$/d$t$ calculated using the 20 data subsets described above. Earlier than 1988, d$(\Delta pCO_2)$/d$t$ was assumed to be the average over the 1988–1993 period. Using annually averaged atmospheric $pCO_2$ values from Mauna Loa and these d$(\Delta pCO_2)$/d$t$, estimates of the water $pCO_2$ are then calculated for each year (Supplementary Figure 3). Results show that shelf water $pCO_2$ could have been lower than atmospheric $pCO_2$ as early as 1950.

**Data availability**. The data that support the findings of this study are available from the corresponding author on reasonable request.

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

## Acknowledgements

G.G.L. was funded as a postdoctoral researcher of FRS-FNRS throughout the execution of this study. We acknowledge the seminal contribution of the Surface Ocean CO$_2$ Atlas (SOCAT) effort to our study. SOCAT is an international effort, endorsed by the International Ocean Carbon Coordination Project (IOCCP), the Surface Ocean Lower Atmosphere Study (SOLAS), and the Integrated Marine Biosphere Research (IMBeR) program, to deliver a uniformly quality-controlled surface ocean CO$_2$ database. The many researchers and funding agencies responsible for the collection of data and quality control are thanked for their contributions to SOCAT. The research leading to these results has received funding from the European Union's Horizon 2020 research and innovation program under the Marie Skłodowska-Curie grant agreement no. 643052 (C-

CASCADES project). Goulven G. Laruelle now works at the UMR 7619 Metis, Sorbonne Universités, UPMC, Univ. Paris 06, CNRS, EPHE, IPSL, Paris, France.

## Author contributions

G.G.L. performed all calculations necessary to this study, following an idea central to the conception of this paper by W.-J.C. P.R. coordinated and participated at all stages of the conception and writing of the paper. G.G.L., W.-J.C., and P.R. wrote the manuscript with input by N.G. All co-authors contributed to specific aspects of the analyses and commented on various versions of the manuscript.

## Additional information

**Competing interests:** The authors declare no competing financial interests.

