## [Peer Review File · Nature Communications]

Reviewers' comments:

Reviewer #1 (Remarks to the Author):

The authors have produced a well written, concise study investigating trends in wintertime pCO₂ of several coastal shelves, identifying evidence for a global growth of air-sea pCO₂ disequilibria. The article will be of interest to the ocean biogeochemical community, and to scientists studying the global climate response to anthropogenic perturbation generally. The paper describes an interesting approach of identifying long term (multiannual to decadal) trends in wintertime coastal sea pCO₂ by excluding data from outside winter months, for which interannual variability in the biologically-driven processes of the carbon cycle obscure long term trends. In this regard, the authors are presenting a novel, important and publishable result, in spite of the limitations of the underlying data (with respect to patchy sampling and relatively short temporal duration). However, there are a couple of points that do require addressing to ensure that the major claims of the paper can be confidently presented before I could recommend the manuscript for publication. Furthermore, although the manuscript appears to me to have been carefully written and proofread, there are a few minor points detailed in the Line-Specific points.

General Points

1) The authors focus their study of trends of coastal ocean pCO₂ on wintertime values rather than use data from over the whole year. This choice is justified by the assertion that summertime pCO₂ values are highly variable due to interannual variability in the competing balance between biological carbon fixation and respiration. The authors suggest that the wintertime sea pCO₂ represents a kind of 'baseline' that responds primarily to the anthropogenic increase of atmospheric pCO₂, and that the inclusion of pCO₂ observations from other times of the year introduces interannual noise (line 116) that can mask out any underlying trend. The authors cite McNeil and Sasse [2016, Nature] to support the claim that winter data "likely reflects the long-term trend better" than annual averages, but it is not clear to me how the cited paper (which investigates how the amplification of the seasonal ocean pCO₂ cycle will expose the surface ocean to high pCO₂ values earlier than would be expected based on purely atmospheric pCO₂ projections) justifies this claim. Further clarification of this important point is necessary.

Indeed, the authors point out in cases where relevant studies have identified the importance of the winter period in controlling the overall annual net CO₂ flux, but it is not totally clear to me that wintertime pCO₂ trends reflect "better the long-term trend in air-sea exchange" (line 184). I agree with the case presented by the authors that coastal wintertime pCO₂ data show a tendency towards increasing $\Delta p\text{CO}_2$ (in spite of the fact that many of the trends' error bounds include values of 0 $\mu\text{atm/yr}$). However, more evidence is required to demonstrate that these wintertime trends are better estimates of the overall annual trends in all cases presented here. My concern is that this approach of only using wintertime data may not capture long term changes which emerge when data from other times of the year are included. Certainly, in many locations, winter is the critical

season in setting the local CO₂ flux, but more evidence must be provided to justify the claim that this is always the case for all locations in this study. Chen and Borges [2009, DSR11], for example, present a collation of estimates of carbon fluxes in the coastal ocean that includes evidence that wintertime pCO₂ is not necessarily primal in all locations, including some locations common to this study (their Table 2, for example in the Gulf of Biscay, the North Sea).

The authors present a convincing case that coastal winter pCO₂ data show evidence of important trends, but to suggest that these winter trends dominate the overall trends requires more evidence (in the form of more references or more firsthand data analysis). This article presents (in terms of Δ pCO₂ trends) an important part of the story by focusing on the winter data, but this must be further stressed: that the winter trends are part of, rather than representative of the whole story.

2) The authors make good use of statistical analysis to present their results, and include a useful amount of detail describing their methods, however I think the manuscript would be improved with a few additional comments. The supplementary material describes the use of F-tests and p values to establish the significance of the trends calculated, but this manuscript would benefit from a comment that the assumptions underlying ordinary least squares regressions are not violated. In time series analyses, it is common that datapoints (more specifically, datapoints' residuals from linear regressions) are serially correlated, i.e. that the variance of a datapoint at a given point in time is determined to a certain extent by the variance of the datapoint at the preceding point in time. Such serial correlation does not affect the size of the regression trends calculated, but it may affect the size of confidence intervals, which can have implications for trends' significance. A comment should be made to satisfy readers that the authors have checked for violations of assumptions of their statistical models and significance tests. In particular, it would be desirable to see a comment to verify that serial correlation is absent/small (which can be determined using tests such as the Durbin-Watson or others) or has been appropriately accounted for (in which case those details should be included). It may be that calculating the trends based on wintertime means (rather than the full annual cycle) adequately diminishes serial correlation that could be otherwise problematic.

I suspect that the trends presented here are representative of the trends that the data show, although the importance of the uncertainties of those trends can hardly be overstated, and I believe the authors have exercised due care in making the caveats of their findings clear. Correctly, the authors note that although these time series are long given their spatial coverage, they are also in practice are too short to reveal decadal trends with a great degree of confidence (lines 119 and 152). They also note that their aggregated global winter dpco₂ trends' error bounds do encompass the 0 mark (243, 244). Therefore, my criticism is not that the authors do not make clear the statistical caveats of their quantitative findings, but instead that they add a little more clarification, to demonstrate to readers that their statistical methods have been applied with due attention given to underlying assumptions.

3) Ensure the manuscript conforms to the 'fair use' requests of openly available datasets (particularly SOCAT, but also check for other data providers' requirements specific to GLOBALVIEW, NCEP/NCAR and any others). Users of SOCAT data, for example, are asked to carefully review and

conform to the data providers' practices for acknowledgement (detailed in the Fair Use Statement http://www.socat.info/SOCAT_fair_data_use_statement.htm which stipulates additional requirements to those listed in Section 2.4.1 of Bakker et al [2014, ESSD]). This asks authors to cite SOCAT publications appropriately (which the authors do well) but also to include SOCAT's acknowledgement statement in the acknowledgement section of the manuscript, as well as notification of publications that use SOCAT data.

Line-specific points:

121: Change the full stop to a comma

124: The paper does well to look at 2 definitions of the coastal margin, and offer justifications for the 'wide' limit and the 20m isobath 'nearshore' limit, but a brief comment to justify the choice of 200m would be useful.

171-173: There needs to be a reference to evidence of the suggested mechanism or a comment to specify that the cause is not determined with anthropogenic nutrient inputs being one possibility. If the thesis cited in the next sentence covers this, consider citing it twice.

182-184: See general point 1).

288: "hindcasts" should be "hindcast"

325: "provides" is an odd word choice: consider using "shows"

426: Check the date of this reference: I believe it should be 2016. Please also ensure that all reference details are correct.

Figures:

The text on all figures is quite small, and only becomes legible when zoomed in substantially.

Figure 2: To make the figure annotations more easily legible, would it be possible to put one large color bar along the top, side or bottom of the entire figure (since all panels use the same scale), leaving more space for the inset histograms to be enlarged?

Supplementary Info

30: Since you define $\Delta pCO_2 = pCO_{2air} - pCO_{2sea}$, state "air pCO_2 " before "water pCO_2 " here.

46: The sentence beginning "However, for 60%..." is difficult to understand. Consider rewording to something like "Of all the cells with a trend in ΔpCO_2 greater than $+0.5 \mu atm/yr$, 60% were considered significant using an F-test with $p < 0.05$ (69% with $p < 0.1$)".

53: Capitalize the reference to Table S2

58: “consisted in” should be “consisted of”

58: “clusters of 20 or more” should be “clusters of 20 or more”

59: Capitalize the reference to Figure S2

118: “a F-test” change to “an F-test”

Reviewer #2 (Remarks to the Author):

The paper aims at determining the trend in $\Delta p\text{CO}_2$ for the “global shelf” using the $p\text{CO}_2$ data available in the SOCAT database.

The authors address a challenge as $p\text{CO}_2$ is highly variable (in time and space) in the coastal ocean (much more than in the open ocean), data are sparse and continental shelves exhibit different behaviors. The long-term trends in the open ocean rely on 30 years of data collected at a few time series stations. On the continental shelf, the time series have a shorter record. The authors propose to use the SOCAT database (introduction) to determine whether the continental shelves show a change in $\Delta p\text{CO}_2$ over time.

The paper does not meet the standards for Nature. It lacks novelty and shows major technical and interpretation problems. Determining long-term trends in such variable regions would require much more data over longer periods.

The abstract is overselling the paper:

- “using a global database covering more than 30 years of observations”: only in a few cases more than 30 years of observations are available.
- The authors extrapolate the results from a few locations to the global shelf. In addition, the map (figure 1) shows a strong bias towards the shelves at temperate latitudes in the Northern hemisphere. This questions the representativeness of their results.
- “increased biological production”: the study focuses on wintertime to avoid biological production, no evidence for increased biological production is shown.
- The two mechanisms suggested (faster exchange of shelf CO_2 and biological production) refer to previous works (in the section “model-derived assessments of the global shelf CO_2 sink” Bauer et al., 2013; Cai, 2011; Andersson and Mackenzie, 2002), no evidence is provided here.

The authors reviewed some previous work regarding local changes in shelf $p\text{CO}_2$ and concluded: “while illustrative, such trends from a handful of locations do not allow drawing any conclusion regarding the overall change in shelf air-sea $p\text{CO}_2$ gradient over time”. Yet, this is what the authors

are doing in this paper. In the section “Global shelf CO₂ sink from winter observations”, they note that the “coverage is largely heterogeneous and partly skewed towards temperate latitudes in the northern hemisphere, this surface area represents ~8% of the global continental shelves”. At the best, they identified 15 regions allowing reconstruction of the evolution of $\Delta p\text{CO}_2$ with a timespan of at least a decade without any region between 30°S and 30°N (except part of SAB near 30°N) and only 3 regions in the Southern hemisphere (figure S2).

The determination of the rate of change in $\Delta p\text{CO}_2$ is not convincing given the high variability in coastal $p\text{CO}_2$. In the SOCAT database, there is a bias in the sampling with more cruises in recent years than in early years. The variance is higher in recent years compared to early years (as observed in figure S3).

In table S2, the standard deviation of the trends is much higher than the rate of change, therefore the sign of the trends is undetermined.

The rate of change is estimated by linear regressions. The normality of residuals, constant variance and linearity of the relationship must be checked for applying the linear regression.

“Results from an analysis performed using all seasonal data are presented in SI, but we report here wintertime data only, when photosynthetic activity is generally the weakest”. The trend in winter is not necessarily the same as the trend in other seasons. In table 1, different seasonal trends are reported by Hauri et al. (2015) for the Antarctic Peninsula. This result illustrates that the extrapolation from winter to other seasons is not valid.

Because of the lack of data, the definition of continental shelf is modified with the boundary relaxed to 500 m or 100 km from the coast. The authors note that the use of the broader definition of the continental shelves affects the calculation of the trend (in section “global shelf CO₂ sink from winter observations”).

The last part of the paper, as noted by the authors, is highly speculative as they assume that the linear trends apply for the period 1950-2010.

Reviewer #3 (Remarks to the Author):

Title: Ocean margins as a variable but increasing sink for atmospheric carbon dioxide

Authors: Laruelle et al

Reviewer: Wiley Evans

Overview:

Laruelle et al construct a global analysis of the time rate of change in the air-sea CO₂ gradient on the continental shelf using the SOCAT V3 dataset. Their results suggest that globally, the air-sea gradient is increasing at a rate of $\sim 1 \text{ uatm yr}^{-1}$, albeit there is considerable variability both across and within shelf regions. The analysis is nicely constructed, the manuscript is well written, and the results highlight two central findings: (1) unlike the open ocean, the trajectory of coastal ocean surface pCO₂ in the majority of regions examined differs from that of the atmosphere, and (2) globally coastal oceans have switched from CO₂ source to CO₂ sink as recently as the 1950's with accelerated atmospheric CO₂ uptake since the late 1990's. These are important findings, and worthy of publication in Nature Communications following reconciliation of my minor comments below.

Minor comments:

Suggest not using ocean margin interchangeably with continental shelf because ocean margin might be more inclusive, containing all areas from the fjord/estuary to the continental slope, whereas your areal definition of continental shelf is more confined.

Using winter data for this analysis is a good idea, and a way, perhaps, to avoid deseasonalizing the data if the full year record were to be used. It doesn't look like the full year records were deseasonalized in the supplemental material. Perhaps a statement in the supplemental material would be good here? The reason this caught my attention is because a significantly difference rate of change is reported here compared to the Bering Sea estimate of Takahashi et al., 2009. I also note that your Bering Sea data are limited to the most southern region near the Aleutian Island with no grid cells over the shelf.

You reference the Oregon shelf as one of the 15 regions, but Figure 2F isn't actually showing a majority of Oregon data. Only the "wide" definition data are actually off the Oregon coast, the "narrow" data are WA and BC. This area, specifically the data dense region near the Strait of Juan de Fuca, has been examined somewhat in the following paper:

Evans, W., B. Hales, P. G. Strutton, and D. Ianson (2012), Sea-air CO₂ fluxes in the western Canadian coastal ocean, Progress in Oceanography, doi: 10.1016/j.pocean.2012.1001.1003.

It may be the size of the squares for the “narrow” definition, but the look to go past the 100 km from shore “wide” delineation. At 50N, 0.5 degrees of longitude is 36 km, and I count 6 squares along the lowest band of cells on the WA state cluster.

It may also be helpful to match the color scale of Figure 1, with yellow centered on zero but with more gradation. There are light green areas in Figure 3 that were yellow in Figure 1.

SUMMARY

We thank the three reviewers for their detailed and helpful comments and suggestions, many of which we implemented in the revised manuscript. The most important comments/concerns were the following: (i) Trends computed from winter-time only data are not representative of the trends for the entire year. (ii) The number of regions with significant positive trends is too small to infer a global trend, and (iii) the statistical methods are not adequate, and lead to biased results.

In response to these major points, we substantially expanded on the methods and data, and also toned down some of the more speculative elements of the paper. Specifically, we now include a trend analysis based on deseasonalized data, which permits us to use data from all seasons. We can show that the main conclusions are not affected by this change. We also altered our main trend computation method, and also include now a large range of algorithms to test for the significance of the trends. Finally, we toned down our extrapolation.

We hope that our changes address these concerns to the full satisfaction of the reviewers.

Please find below a detailed answer to each comment by the three reviewers. All our answers are written in blue and the questions and comments of the reviewers are left in black. All the sections of the manuscript that have been updated are indicated in red and bold.

On behalf of all co-authors,

Goulven Laruelle

Reviewer #1 (Remarks to the Author):

The authors have produced a well written, concise study investigating trends in wintertime pCO₂ of several coastal shelves, identifying evidence for a global growth of air-sea pCO₂ disequilibria. The article will be of interest to the ocean biogeochemical community, and to scientists studying the global climate response to anthropogenic perturbation generally. The paper describes an interesting approach of identifying long term (multiannual to decadal) trends in wintertime coastal sea pCO₂ by excluding data from outside winter months, for which interannual variability in the biologically-driven processes of the carbon cycle obscure long term trends. In this regard, the authors are presenting a novel, important and publishable result, in spite of the limitations of the underlying data (with respect to patchy sampling and relatively short temporal duration). However, there are a couple of points that do require addressing to ensure that the major claims of the paper can be confidently presented before I could recommend the manuscript for publication. Furthermore, although the manuscript appears to me to have been carefully written and proofread, there are a few minor points detailed in the Line-Specific points.

We are grateful for the reviewer's evaluation and his/her constructive suggestions.

General Points

1) The authors focus their study of trends of coastal ocean pCO₂ on wintertime values rather than use data from over the whole year. This choice is justified by the assertion that summertime pCO₂ values are highly variable due to interannual variability in the competing balance between biological carbon fixation and respiration. The authors suggest that the wintertime sea pCO₂ represents a kind of 'baseline' that responds primarily to the anthropogenic increase of atmospheric pCO₂, and that the inclusion of pCO₂ observations from other times of the year introduces interannual noise (line 116) that can mask out any underlying trend. The authors cite McNeil and Sasse [2016, Nature] to support the claim that winter data "likely reflects the long-term trend better" than annual averages, but it is not clear to me how the cited paper (which investigates how the amplification of the seasonal ocean pCO₂ cycle will expose the surface ocean to high pCO₂ values earlier than would be expected based on purely atmospheric pCO₂ projections) justifies this claim. Further clarification of this important point is necessary.

[R1A1] We thank the referee for his/her general approval of our approach to identify the decadal trend using wintertime data and for the suggestion that further clarification is needed to strengthen the paper. In addition to our arguments given at the beginning of the Methodology, we provide below a few more justifications.

First, to address the concerns of the referee and take into account his/her suggestion, we now present an analysis of the SOCAT dataset for all seasons using a deseasonalized approach based on a recently published neural network approach (Laruelle et al. 2017) (see **A1R3** for more details). The results are not greatly different from those obtained using the wintertime data only. In the revised manuscript, the "all year" results are presented along with the winter only data for the global analysis (Lines 268-279). However, we believe that despite all the advantages of the neural network approach to perform the deseasonalization, some seasonal signals cannot totally be removed. The results using year-long data are thus significantly noisier than those obtained using winter data only

Secondly, the paper by McNeil and Sasse (2016) does not explicitly state that winter data "likely reflects the long-term trend better" than annual averages. However, it contains the following statement: "Although there were twice as many global summertime measurements for carbon as there were for winter, we found no seasonal bias from our independent testing of SOMLO." This suggests that their data do not show significant differences between the trends observed for winter and for the whole year. In our manuscript, the reference to McNeil and Sasse (2016) was introduced after stating the fact that, in our data, the whole year trends exhibit much larger variability than those calculated using only winter data. We thus asserted that the use of winter data "likely reflect the long term trend better". Therefore, the reference to McNeil and Sasse (2016) was not correct in this context and has been removed in the updated manuscript.

Third, data collected in spring, summer and fall are known to be affected by biological and other processes. For example, in the East China Sea Chou et al. (2011) reported that due to increased eutrophication in the past decades, surface water pCO₂ is becoming lower in spring and higher in the fall. Another support against the use a summer data for trend analysis comes from the North Sea where two sets of summertime data (2005 and 2001) revealed that water pCO₂ increased at a rate 5 times faster than the atmosphere (Thomas et al., 2007). A later study, however, using summer data from years 2001, 2005 and 2008 revealed a large increase of 26 μatm between 2001 and 2005 consistent with the earlier results of Thomas et al., 2007 but only a moderate increase of 4 μatm

between 2005 and 2008 (Salt et al., 2013). The main manuscript explicitly refers to this example (lines 183-187). We thus believe that using winter data only and focusing on time series with measurements in at least 4 different years is a valid approach to deciphering the long-term dynamics of the air-water pCO₂ gradient in continental shelf seas.

Finally, we emphasize (though we do not feel the referee has any misunderstanding on this issue) that while we use wintertime data to examine the rate of change (trend), we do not assume that the annual air-sea CO₂ flux is the same as wintertime flux.

Indeed, the authors point out in cases where relevant studies have identified the importance of the winter period in controlling the overall annual net CO₂ flux, but it is not totally clear to me that wintertime pCO₂ trends reflect “better the long-term trend in air-sea exchange” (line 184). I agree with the case presented by the authors that coastal wintertime pCO₂ data show a tendency towards increasing $\Delta p\text{CO}_2$ (in spite of the fact that many of the trends’ error bounds include values of 0 $\mu\text{atm}/\text{yr}$). However, more evidence is required to demonstrate that these wintertime trends are better estimates of the overall annual trends in all cases presented here. My concern is that this approach of only using wintertime data may not capture long term changes which emerge when data from other times of the year are included. Certainly, in many locations, winter is the critical season in setting the local CO₂ flux, but more evidence must be provided to justify the claim that this is always the case for all locations in this study. Chen and Borges [2009, DSR1], for example, present a collation of estimates of carbon fluxes in the coastal ocean that includes evidence that wintertime pCO₂ is not necessarily primal in all locations, including some locations common to this study (their Table 2, for example in the Gulf of Biscay, the North Sea).

[R1A2] We understand the reviewer’s concern regarding the importance of the winter season in the overall annual dynamics of the coastal CO₂ exchange at the air-water interface. Our main point, however, is not to claim that winter contributes the most to the coastal carbon sink but that the variability, in particular inter-annual, is less important in winter than over the entire year or during any other season, which justifies our choice to work with winter data alone. This has been clarified in the updated manuscript (lines 120-121).

“This does not suggest however that winter contributes more than other seasons to the overall annual trend.”

Please see also our answer **[R1A1]**.

The authors present a convincing case that coastal winter pCO₂ data show evidence of important trends, but to suggest that these winter trends dominate the overall trends requires more evidence (in the form of more references or more firsthand data analysis). This article presents (in terms of $\Delta p\text{CO}_2$ trends) an important part of the story by focusing on the winter data, but this must be further stressed: that the winter trends are part of, rather than representative of the whole story.

[R1A3] We agree with the reviewer that the representativeness of the trend observed in winter is a critical question that needs to be further investigated in our study. In the original version of the manuscript, table S1 compared the number of cells for which negative ($< -0.5 \mu\text{atm yr}^{-1}$), neutral and

positive rates of increase in $\Delta p\text{CO}_2$ ($>+0.5 \mu\text{atm yr}^{-1}$) were calculated using only winter data and using data from the entire year. The percentage of cells for which the trend was statistical significant was also reported for each category (i.e. negative, neutral or positive). These results were only briefly mentioned in the main text and are now further elaborated and discussed in the revised manuscript.

As demonstrated by the figures below, the vast majority of the cells for which a statistically significant trend is detected (using an F-test with $p < 0.05$) display positive values corresponding to an increase of the winter $\Delta p\text{CO}_2$ over time (Figure 1, left panel, red bars). The same calculations were then performed using data for the entire year after deseasonalization using the 0.25° monthly $p\text{CO}_2$ climatology for the continental shelf seas reported in Laruelle et al. (2017). The all-season results are presented in Figure 1, right panel. Although less pronounced, the same tendency is observed with positive values for the majority of cells for which a statistically significant trend was observed (red bars). This result supports our hypothesis that the trends calculated for winter are, at least qualitatively, representative of a trend taking place over the entire year. In the revised manuscript, we have thus added (lines 273-279):

“Although much noisier, the overall annual $d(\Delta p\text{CO}_2)/dt$ reveal qualitatively similar trends to those observed with data from the winter months only (Table S1). The overall proportion of cells displaying statistically significant trends is much lower when all year data are used (22%) than when winter data are retained (45%). Nevertheless, nearly three times more cells display significant trends for which $d(\Delta p\text{CO}_2)/dt > +0.5 \mu\text{atm yr}^{-1}$ (574) than for which $d(\Delta p\text{CO}_2)/dt < -0.5 \mu\text{atm yr}^{-1}$ (201), a result in broad agreement with our winter analysis only. Therefore, the analysis of deseasonalized data for all seasons also point toward a tendency for an enhanced shelf uptake of atmospheric CO_2 .”

Figure 1: Distribution of $d(\Delta p\text{CO}_2)/dt$ in winter (left) and during the whole year (right) for all cells (black) and cells for which a statistically significant trend could be established using a F-test with $p < 0.05$ (red). The all-year data were deseasonalized using the 0.25° monthly $p\text{CO}_2$ climatology of Laruelle et al. (2017).

In the updated manuscript, the results presented in the figures above are included in the main text as well as in Figure 1 and more attention has been drawn to the results presented in table S1 which compares trends calculated using winter data against trends calculated using deseasonalized data from the entire year (lines 268-279). We acknowledge however, that the trend observed in winter

does not reflect the entirety of the signal observed for the whole year but that it can be used as a qualitative proxy.

2) The authors make good use of statistical analysis to present their results, and include a useful amount of detail describing their methods, however I think the manuscript would be improved with a few additional comments. The supplementary material describes the use of F-tests and p values to establish the significance of the trends calculated, but this manuscript would benefit from a comment that the assumptions underlying ordinary least squares regressions are not violated. In time series analyses, it is common that datapoints (more specifically, datapoints' residuals from linear regressions) are serially correlated, i.e. that the variance of a datapoint at a given point in time is determined to a certain extent by the variance of the datapoint at the preceding point in time. Such serial correlation does not affect the size of the regression trends calculated, but it may affect the size of confidence intervals, which can have implications for trends' significance. A comment should be made to satisfy readers that the authors have checked for violations of assumptions of their statistical models and significance tests. In particular, it would be desirable to see a comment to verify that serial correlation is absent/small (which can be determined using tests such as the Durbin-Watson or others) or has been appropriately accounted for (in which case those details should be included). It may be that calculating the trends based on wintertime means (rather than the full annual cycle) adequately diminishes serial correlation that could be otherwise problematic.

[R1A4] We agree with the reviewer's comment that linear regressions and the use of the ordinary least square method involves a number of assumptions that need to be checked in order to use the results. In the updated manuscript and supplementary information, we now provide the reader with the necessary information regarding the applicability of our statistical treatment.

First, the normality of the distribution of the residual has been tested using a Kolmogorov and Smirnov test and over 90% of the times series do not reveal significant biases compared to a normal distribution. Second, as suggested by the reviewer, the Durbin Watson test has been used on all time-series and yields an average s of 2, indicating that no systematic auto-correlation is observed in the data. Furthermore, the test rejects the null hypothesis (that some auto-correlation exists in the data) for less than 10% of the time series.

In addition, these tests were also used to more objectively determine the minimum size of the data series for which a trend could be calculated. In the revised manuscript, the thresholds are now set to at least 10 data points over a minimum of 5 different years within a time span of at least 10 years. The selection of these thresholds is now explicitly discussed in the updated Supplementary Information.

"Prior to calculating the linear regressions, the following statistical tests were performed for each cell:

1. The normality of the distribution of the residuals has been evaluated using a Kolmogorov and Smirnov test and >95% of the time series do not exhibit significant deviations from a normal distribution (figure S2a)

2. The existence of auto correlation within the time series has been diagnosed using a Durbin-Watson's test. Results yield values clustered around 2, revealing no significant auto-correlation (figure S2b)

3. The consistency of the variance of the residuals over time was assessed using White's test and significant heteroscedasticity was detected in about half of the times series. As a consequence, the linear regressions were calculated using the bi-square method rather than the simple least square method, which is less robust when the residual is not consistent throughout the entire dataset.

These statistical tests were also used to determine the minimum required length of the time series (i.e. data spanning at least ten years, more than ten individual data, at least five different years with data). Using this set of criteria, the average length of our time series is 18 years."

I suspect that the trends presented here are representative of the trends that the data show, although the importance of the uncertainties of those trends can hardly be overstated, and I believe the authors have exercised due care in making the caveats of their findings clear. Correctly, the authors note that although these time series are long given their spatial coverage, they are also in practice are too short to reveal decadal trends with a great degree of confidence (lines 119 and 152). They also note that their aggregated global winter dpco2 trends' error bounds do encompass the 0 mark (243, 244). Therefore, my criticism is not that the authors do not make clear the statistical caveats of their quantitative findings, but instead that they add a little more clarification, to demonstrate to readers that their statistical methods have been applied with due attention given to underlying assumptions.

[R1A5] We thank the reviewer for acknowledging our effort to perform a sound statistical analysis and bring up its potential limitations. Following the reviewer's advice we now make sure that the updated manuscript clarifies the context in which this analysis is performed and its underlying assumptions (see answers **[R1A4]** and **[R2A10]**). However, regarding the fact that the aggregated global winter ΔpCO_2 trends' error bounds do encompass the 0 mark, this should be interpreted as a sign that the global trend is not significant. It means that the increase in CO_2 sink over time is not significant everywhere in the world but the distribution of the statistically significant trends clearly suggests that the vast majority of regions display a positive trend.

3) Ensure the manuscript conforms to the 'fair use' requests of openly available datasets (particularly SOCAT, but also check for other data providers' requirements specific to GLOBALVIEW, NCEP/NCAR and any others). Users of SOCAT data, for example, are asked to carefully review and conform to the data providers' practices for acknowledgement (detailed in the Fair Use Statement

http://www.socat.info/SOCAT_fair_data_use_statement.htm [3] which stipulates additional requirements to those listed in Section 2.4.1 of Bakker et al [2014, ESSD]). This asks authors to cite SOCAT publications appropriately (which the authors do well) but also to include SOCAT's acknowledgement statement in the acknowledgement section of the manuscript, as well as notification of publications that use SOCAT data.

[R1A6] The reviewer is correct, our previous manuscript did not contain any acknowledgments section upon initial submission but we now made sure to include one which appropriately refers to the SOCAT initiative and the data holders (Lines 326-331).

"Acknowledgments

'The Surface Ocean CO₂ Atlas (SOCAT) is an international effort, endorsed by the International Ocean Carbon Coordination Project (IOCCP), the Surface Ocean Lower Atmosphere Study (SOLAS) and the Integrated Marine Biosphere Research (IMBeR) program, to deliver a uniformly quality-controlled surface ocean CO₂ database. The many researchers and funding agencies responsible for the collection of data and quality control are thanked for their contributions to SOCAT.'

Line-specific points:

121: Change the full stop to a comma

[R1A7] The text has been updated as suggested by the reviewer.

124: The paper does well to look at 2 definitions of the coastal margin, and offer justifications for the 'wide' limit and the 20m isobath 'nearshore' limit, but a brief comment to justify the choice of 200m would be useful.

[R1A8] We thank the reviewer for acknowledging the usefulness of setting two different outer boundaries for the continental shelf. The rationale for using the 200m isobath comes for the wide spread use of this limit in the literature since Walsh et al. (1998), especially for several global estimates of the global coastal C sink (Borges, 2005; Borges et al., 2005, Chen and Borges, 2009; Laruelle et al., 2010; Cai, 2011...).

The following sentence was added to the manuscript (Lines 123-124):

"This depth is commonly used in the literature as the depth at which the shelf breaks (Walsh, 1988; Borges et al., 2005; Laruelle et al., 2010)."

171-173: There needs to be a reference to evidence of the suggested mechanism or a comment to specify that the cause is not determined with anthropogenic nutrient inputs being one possibility. If the thesis cited in the next sentence covers this, consider citing it twice.

[R1A9] We now made sure to cite Wesslander's work earlier in the updated manuscript when referring to the mechanism suggested to influence the CO₂ dynamics in the Baltic Sea (Line 178).

182-184: See general point 1).

[R1A10] See answer [R1A1].

288: "hindcasts" should be "hindcast"

[R1A11] In the sentence the reviewer is referring to, 'hindcasts' does not suggest a plural but is used as a verb and the 's' is only there to make the verb agree with the subject (i.e. 'A recent model').

325: "provides" is an odd word choice: consider using "shows"

[R1A12] The text has been updated as suggested by the reviewer.

426: Check the date of this reference: I believe it should be 2016. Please also ensure that all reference details are correct.

[R1A13] Indeed, the reviewer is correct and the reference is from 2016. All citations and references will be thoroughly checked before resubmission.

Figures:

The text on all figures is quite small, and only becomes legible when zoomed in substantially.

[R1A14] We tried to improve the readability of the figures by increasing the font of the legends.

Also, on figure 1 (now figure 2 in the updated manuscript), we extracted both inside panels to make them larger and, on figure 2 (now figure 1 in the updated manuscript), we now only insert 1 internal histogram panel in each map, thus allowing each histogram to be larger and easier to read.

Revised Figure 2

Figure 2: To make the figure annotations more easily legible, would it be possible to put one large color bar along the top, side or bottom of the entire figure (since all panels use the same scale), leaving more space for the inset histograms to be enlarged?

[R1A15] The panels of the figure have been modified as requested by the reviewer and the scale was placed at the bottom of the figure.

Revised Figure 1

Supplementary Info

30: Since you define $\Delta p\text{CO}_2 = p\text{CO}_2\text{air} - p\text{CO}_2\text{sea}$, state “air $p\text{CO}_2$ ” before “water $p\text{CO}_2$ ” here.

[R1A16] The text has been updated as suggested by the reviewer.

46: The sentence beginning “However, for 60%...” is difficult to understand. Consider rewording to something like “Of all the cells with a trend in $\Delta p\text{CO}_2$ greater than $+0.5 \mu\text{atm/yr}$, 60% were

considered significant using an F-test with $p < 0.05$ (69% with $p < 0.1$)”.

[R1A17] We agree with the reviewer that this phrasing was misleading. In the updated manuscript, more attention has been paid to those results (see answer [R1A3]) and this section has been rewritten as follows.

“In the majority of the cells characterized by a small rate of change in $\Delta p\text{CO}_2$ (27 out of 246), the trends revealed by the linear regression are not statistically significant because the inter-annual variability overshadows any long-term trend. However, out of 495 cells displaying an increase in $\Delta p\text{CO}_2 > +0.5 \mu\text{atm yr}^{-1}$, 325 (66%) are considered statistically significant using an F-test with $p < 0.05$ (71% significance with $p < 0.1$). On the other hand, among the 84 cells for which $d(\Delta p\text{CO}_2)/dt < -0.5 \mu\text{atm yr}^{-1}$, only 22 (26%) display significant trends with $p < 0.05$.”

53: Capitalize the reference to Table S2

[R1A18] The text has been updated as suggested by the reviewer.

58: “consisted in” should be “consisted of”

[R1A19] The text has been updated as suggested by the reviewer.

58: “clusters of 20 of more” should be “clusters of 20 or more”

[R1A20] The text has been updated as suggested by the reviewer.

59: Capitalize the reference to Figure S2

[R1A21] The text has been updated as suggested by the reviewer.

118: “a F-test” change to “an F-test”

[R1A22] The text has been updated as suggested by the reviewer.

Reviewer #2 (Remarks to the Author):

The paper aims at determining the trend in $\Delta p\text{CO}_2$ for the “global shelf” using the $p\text{CO}_2$ data available in the SOCAT database. The authors address a challenge as $p\text{CO}_2$ is highly variable (in time and space) in the coastal ocean (much more than in the open ocean), data are sparse and continental shelves exhibit different behaviors. The long-term trends in the open ocean rely on 30 years of data collected at a few time series stations. On the continental shelf, the time series have a shorter record. The authors propose to use the SOCAT database (introduction) to determine whether the continental shelves show a change in $\Delta p\text{CO}_2$ over time.

The paper does not meet the standards for Nature. It lacks novelty and shows major technical and interpretation problems. Determining long-term trends in such variable regions would require much more data over longer periods.

[R2A1] The SOCAT database is the largest oceanic $p\text{CO}_2$ database to date. We certainly agree with the reviewer that the data distribution is heterogeneous and is skewed toward northern latitude regions, but we would also like to point out that over 40% of the data contained in the SOCAT database are coastal (Bakker et al., 2015) and a significant effort has been paid in recent years to better represent nearshore regions. It might take several years or even decades before the data coverage over the continental shelf is truly representative of the entire globe both in time and space but, we believe the current coverage, which amounts to ~6% of the global coastal ocean is sufficient to decipher significant trends and already exceeds the coverage of the open ocean several years ago when decadal trends of the global oceanic CO_2 sink were first investigated (Takahashi et al., 2003; 2006; 2009; Le Quéré et al, 2010).

While we are fully aware of the limitations, we do not agree with the referee that one has to wait until more data are available to address the sea surface $p\text{CO}_2$ trend in the continental shelves. Numerous papers have been published in the past two decades on the analyses of open ocean trends, despite obvious data limitations. We consider it important to push the analysis forward, such that we can start recognizing trends early on. Waiting until 100% confidence is reached to carry out such an analysis will stifle progress. We believe that the data are sufficient to examine the decadal trend on the rate of sea surface $p\text{CO}_2$ change on continental shelves and our effort here will stimulate more field data collection and motivate further data analysis using improved methods (for example the deseasonalized trend analysis over data from all season as is preliminary done in the new figure 3).

The abstract is overselling the paper:

[R2A2] In the updated version of the manuscript, the abstract was toned down.

“The partial pressure of carbon dioxide ($p\text{CO}_2$) in the open ocean has tended to increase in sync with the rise in atmospheric CO_2 ^{1,2,3}, though regional and temporal differences have been observed⁴⁻⁷. However, previous researchers speculated that the $p\text{CO}_2$ increase on the continental shelves may not follow this open ocean pattern⁸⁻¹⁰. Using a global database spanning a period of up to 35 years¹¹, we

show that although highly variable in different shelf regions, the rates of sea surface pCO₂ increases tend to be smaller than that of the atmosphere. **Linear regressions were calculated in half degree resolution cells using both wintertime data only and deseasonalized annual data. In the case of the wintertime analysis, the air-sea pCO₂ gradient (ΔpCO_2) increased in 653 of the 825 cells for which a trend could be calculated. In 325 of these cells, the increase in ΔpCO_2 is faster than $+0.5 \mu atm yr^{-1}$ and statistically significant (F-test, $p < 0.05$). On the other hand, only 22 cells display a statistically significant decrease in ΔpCO_2 faster than $-0.5 \mu atm yr^{-1}$.**

Although noisier, the deseasonalized annual data suggest similar results with three times more cells displaying statistically significant increase in $\Delta pCO_2 > +0.5 \mu atm yr^{-1}$ than a statistically significant decrease in $\Delta pCO_2 < -0.5 \mu atm yr^{-1}$. Both cases point toward a tendency for an enhanced shelf uptake of atmospheric CO₂. If this was a global trend, it would lend support to the idea that shelves have switched from a source to a sink of CO₂ during the course of the last century.

- "using a global database covering more than 30 years of observations": only in a few cases more than 30 years of observations are available.

[R2A3] It is true that only a limited number of times series span as much as 30 years in the SOCAT database. In the updated manuscript, we have now modified the previous statement (Line 17) by stating:

"Using a global database spanning a period of up to 35 years, ..."

In addition, we now also mention the average length of the time series used in our analysis (18 years) in the SI.

"Using this set of criteria, the average length of our time series is 18 years."

As well as in the main text (Line 166):

"Our analysis employing a narrow definition of the continental shelf provides decadal trends in ΔpCO_2 , i.e., $d\Delta pCO_2/dt$ values, for 825 cells with an average length of our time series of 18 years."

- The authors extrapolate the results from a few locations to the global shelf. In addition, the map (figure 1) shows a strong bias towards the shelves at temperate latitudes in the Northern hemisphere. This questions the representativeness of their results.

[R2A4] We agree with the reviewer (and regret) that data are not available in more regions. The reviewer is also right to point out that a bias towards Northern temperate shelves exists in our analysis, but it is a consequence of the current bias in sampling efforts. We do acknowledge this geographic bias in lines 246-249 of the revised manuscript. We are now also more restrained in our extrapolation to the global shelf.

"While this coverage is largely heterogeneous and partly skewed towards temperate latitudes in the northern hemisphere, this surface area represents ~6% of the global continental shelves and the bulk of our winter data consistently show trends that are dominantly in accordance with an increase of the CO₂ sink on the continental shelf."

- “increased biological production”: the study focuses on wintertime to avoid biological production, no evidence for increased biological production is shown.

[R2A5] Yes, we focus on the wintertime data to avoid interference by inter-annual differences in major biological production. This doesn't mean there is no biological production during winter season and now pay attention in the revised manuscript not to suggest such statement.

- The two mechanisms suggested (faster exchange of shelf CO₂ and biological production) refer to previous works (in the section “model-derived assessments of the global shelf CO₂ sink” Bauer et al., 2013; Cai, 2011; Andersson and Mackenzie, 2002), no evidence is provided here.

[R2A6] The purpose of our manuscript is to analyse the trend we can detect in the available data but we do not pretend to be able to identify the mechanism responsible for these trends. We thus refer to previously published literature to mention known or proposed mechanisms that may affect the dynamics of the CO₂ air-sea exchange over continental shelves for context but do not attempt to demonstrate their existence in our study. We agree that we do not have the possibility to positively identify which mechanism (or combination of mechanisms) drives the dynamics of CO₂ exchange with the atmosphere in continental shelf seas from the data alone but believe that referring to the processes that could be responsible for the trends we observe is important.

The authors reviewed some previous work regarding local changes in shelf pCO₂ and concluded: “while illustrative, such trends from a handful of locations do not allow drawing any conclusion regarding the overall change in shelf air-sea pCO₂ gradient over time”. Yet, this is what the authors are doing in this paper. In the section “Global shelf CO₂ sink from winter observations”, they note that the “coverage is largely heterogeneous and partly skewed towards temperate latitudes in the northern hemisphere, this surface area represents ~8% of the global continental shelves”. At the best, they identified 15 regions allowing reconstruction of the evolution of ΔpCO₂ with a timespan of at least a decade without any region between 30°S and 30°N (except part of SAB near 30°N) and only 3 regions in the Southern hemisphere (figure S2).

[R2A7] First, we must clarify that by saying “a handful of locations” we do not mean that previous analyses and conclusions were drawn based on analysis of data from “a handful of locations”; rather, most prior research used data from one location or limited scale region only and often employed inconsistent approaches. Collectively, the conclusions from these “handful of locations” are contrasting and contradictory. Therefore, we believe our analysis is the first global synthesis and represent a large improvement over previous work. Further, considering the amount of data used in our analysis, we do not agree with the reviewer that it is a mere collection of regional analysis from which no global hypothesis can be inferred. We do, however, agree that the lack of observation along the equatorial band limits the global reach of our conclusions and this is now reflected upon in the revised manuscript (line 249).

“The current global data coverage only contains very few equatorial data however.”

We believe that our results clearly illustrate that in many regions of the world, the coastal sink of CO₂ either exhibits an increase, but we do not claim that the entire coastal ocean homogeneously behaves as an increasing CO₂ sink. We believe that we fairly report the spatial heterogeneity in the calculated trends, but also believe that because 86% of the cells for which a statistically significant trend can be identified reveal an increase of the dpCO₂ over time, we have here a result of global significance (325 out of 374 using p<0.05, see table S1).

Taking into consideration the referee's concern and suggestions, the updated manuscript better conveys this point that not all the continental shelves are characterized by an increasing CO₂ sink, but that the majority of the regions having a change display an increase of the pCO₂ gradient over time. We also believe that our additional approach using deseasonalized data (presented in the supplement) further strengthens our position. Indeed, using deseasonalized data from the entire year, 71% of the cells for which a statistically significant trend could be calculated correspond to an increase of dpCO₂ over time (574 out of 825, see table S1). This is now discussed both in the main text and in the supplementary information:

"A similar analysis was performed using data from all months and evidences that the trends observed in the winter months only are more significant than those calculated for the entire year (Table S1). However, these calculations reveal that, nearly three times more cells display significant trends for which $d(\Delta pCO_2)/dt > +0.5 \mu atm yr^{-1}$ (574) than for which $d(\Delta pCO_2)/dt < -0.5 \mu atm yr^{-1}$ (201). Thus, although much noisier, trends calculated using data from the entire year also suggest that a larger fraction of the continental shelves are characterized by a statistically significant increase in ΔpCO_2 over time than by a decrease in ΔpCO_2 ."

and the main manuscript (lines 268-279):

"The results from the analysis using data across the entire year generally confirm the results from the winter-time only data as exemplified by the change in ΔpCO_2 with time for all cells pertaining to the 6 largest regions used in the regional analysis (Figure 3). For all regions, the range of values of ΔpCO_2 observed in winter (red) is largely less than that observed over the entire year (black). Globally, calculations performed using deseasonalized data from the entire year allow deriving trends for 3721 cells (Table S1). Although much noisier, the overall annual $d\Delta pCO_2/dt$ reveal qualitatively similar trends to those observed with data from the winter months only (Table S1). The overall proportion of cells displaying statistically significant trends is much lower when all year data are used (22%) than when winter data are retained (45%). Nevertheless, nearly three times more cells display significant trends for which $d(\Delta pCO_2)/dt > +0.5 \mu atm yr^{-1}$ (574) than for which $d(\Delta pCO_2)/dt < -0.5 \mu atm yr^{-1}$ (201), a result in broad agreement with our winter analysis only. Therefore, the analysis of deseasonalized data for all seasons also point toward a tendency for an enhanced shelf uptake of atmospheric CO₂."

The determination of the rate of change in ΔpCO_2 is not convincing given the high variability in coastal pCO₂. In the SOCAT database, there is a bias in the sampling with more cruises in recent years than in early years. The variance is higher in recent years compared to early years (as observed in figure S3).

[R2A8] We agree with the reviewer that the past few years are overrepresented in the SOCAT database and this can indeed introduce a bias in the variance over time. In order to quantify this potential bias, the homoscedasticity of the variance of the residuals has been assessed using a White test, which revealed some heteroscedasticity in the residuals of ~50% of the times series.

We thank the referee for pushing us hard to be more rigorous regarding how we derive our conclusions. As a consequence, a more robust method is now used to calculate the coefficients of the linear regressions and the 'ordinary least square method' was replaced by the 'bisquare method', which allows deriving the coefficient of a linear regression with the presence of some

heteroscedasticity or large outliers in the residuals. This change in procedure, however, did not significantly affect the results in terms of magnitude of the trends observed.

In table S2, the standard deviation of the trends is much higher than the rate of change, therefore the sign of the trends is undetermined.

[R2A9] see answer [R1A5].

The rate of change is estimated by linear regressions. The normality of residuals, constant variance and linearity of the relationship must be checked for applying the linear regression.

[R2A10] The reviewer is correct. Several statistical tests must be performed prior to perform linear regressions and the appropriate analysis is now presented in the Supplementary Information. It mentions the following points:

- 1- The normality of the distribution of the residuals has been evaluated using a Kolmogorov and Smirnov test and >95% of the time series do not exhibit significant deviations from a normal distribution (see answer [R1A4])
- 2- The consistency of the variance (i.e. homoscedasticity) has been assessed using White's test and some heteroscedasticity was detected. As a consequence, the calculation method was updated from the 'ordinary least square' to the 'bisquare' method (see answer [R2A8]).
- 3- The existence of auto-correlation has been diagnosed using a Durbin-Watson's test and significant ($p < 0.05$) in less than 10% of the time series. (see answer [R1A5])

"Prior to calculating the linear regressions, the following statistical tests were performed for each cell:

1. The normality of the distribution of the residuals has been evaluated using a Kolmogorov and Smirnov test and >95% of the time series do not exhibit significant deviations from a normal distribution (figure S2a)

2. The existence of auto correlation within the time series has been diagnosed using a Durbin-Watson's test. Results yield values clustered around 2, revealing no significant auto-correlation (figure S2b)

3. The consistency of the variance of the residuals over time was assessed using White's test and significant heteroscedasticity was detected in about half of the times series. As a consequence, the linear regressions were calculated using the bi-square method rather than the simple least square method, which is less robust when the residual is not consistent throughout the entire dataset.

These statistical tests were also used to determine the minimum required length of the time series (i.e. data spanning at least ten years, more than ten individual data, at least five different years with data). Using this set of criteria, the average length of our time series is 18 years."

"Results from an analysis performed using all seasonal data are presented in SI, but we report here wintertime data only, when photosynthetic activity is generally the weakest". The trend in winter is not necessarily the same as the trend in other seasons. In table 1, different seasonal trends are reported by Hauri et al. (2015) for the Antarctic Peninsula. This result illustrates that the extrapolation from winter to other seasons is not valid.

[R2A11] We agree with the reviewer that the trends observed for different seasons may vary as reported by Hauri et al. (2015) for the Antarctic Peninsula. This may partly be the result from a recent tendency towards more pronounced seasonal pCO₂ cycles in the ocean. However, we do believe that the trends derived for winter do qualitatively represent the year-long trend as demonstrated in answer [R1A3]. In addition, as demonstrated in answer [R1A3] and [R2A7], calculations performed using deseasonalized data collected over the entire year do support the trends suggested by the winter data alone.

Because of the lack of data, the definition of continental shelf is modified with the boundary relaxed to 500 m or 100 km from the coast.

The authors note that the use of the broader definition of the continental shelves affects the calculation of the trend (in section “global shelf CO₂ sink from winter observations”).

[R2A12] Indeed, the use of a broader definition of the continental shelf illustrates that the increase of the continental shelf CO₂ sink appears to be more pronounced in shallower region, thus evidencing a coastal process. This makes sense, as the broader definition will include more areas from the open ocean (which has on average no increase in air-sea pCO₂ gradient with time). We believe that using two definitions of the continental shelf (as well as performing our calculations once over the entire ocean) allows us to bring this point across.

The last part of the paper, as noted by the authors, is highly speculative as they assume that the linear trends apply for the period 1950-2010.

[R2A13] We agree with the referee that the last part of the paper is more speculative than the rest of the manuscript. However, we do feel that it is beneficial to put the observed recent trend into the perspective of a longer time frame and loop back to the hypothesis proposed earlier (Mackenzie et al., 2011; Bauer et al. 2013) in the section called ‘**Model-derived assessments of the global shelf CO₂ sink**’. We believe this grand picture will stimulate more research in this direction. While we prefer to keep this part, we admit it is highly speculative and further softened our tone in the revised version of our manuscript (Lines 298-303 & 310-313).

“This approach allows comparing trends derived from observations alone with earlier modelling work (Anderson et al., 2004; Mackenzie et al., 2011). Our calculations suggest that the continental shelves might have turned from a purported pre-industrial source of CO₂ into a sink for atmospheric CO₂ as early as in the 1950s, at least during wintertime which would be consistent with those model results¹⁰ although our data-based estimate may indicate that the switchover time could have occurred earlier than previously thought.”

“Nevertheless, the purpose of this work is to draw the attention to the fact that the rate of increase of the air-sea pCO₂ gradient in continental shelf waters and its importance in global ocean CO₂ uptake is still poorly understood and deserve further study.”

Reviewer #3 (Remarks to the Author):

Title: Ocean margins as a variable but increasing sink for atmospheric carbon dioxide

Authors: Laruelle et al

Reviewer: Wiley Evans

Overview:

Laruelle et al construct a global analysis of the time rate of change in the air-sea CO₂ gradient on the continental shelf using the SOCAT V3 dataset. Their results suggest that globally, the air-sea gradient is increasing at a rate of $\sim 1 \text{ uatm yr}^{-1}$, albeit there is considerable variability both across and within shelf regions. The analysis is nicely constructed, the manuscript is well written, and the results highlight two central findings: (1) unlike the open ocean, the trajectory of coastal ocean surface pCO₂ in the majority of regions examined differs from that of the atmosphere, and (2) globally coastal oceans have switched from CO₂ source to CO₂ sink as recently as the 1950's with accelerated atmospheric CO₂ uptake since the late 1990's. These are important findings, and worthy of publication in Nature Communications following reconciliation of my minor comments below.

Minor comments:

Suggest not using ocean margin interchangeably with continental shelf because ocean margin might be more inclusive, containing all areas from the fjord/estuary to the continental slope, whereas your areal definition of continental shelf is more confined.

[R3A1] We thank the reviewer for noting this potential source of confusion. In response, special attention has been paid in the updated manuscript to strictly refer to the studied domain as 'continental shelf'. As a consequence, we also modified the title from '**Ocean margins** as a variable but increasing sink for atmospheric carbon dioxide' to '**Continental shelves** as a variable but increasing sink for atmospheric carbon dioxide'

Using winter data for this analysis is a good idea, and a way, perhaps, to avoid deseasonalizing the data if the full year record were to be used. It doesn't look like the full year records were deseasonalized in the supplemental material. Perhaps a statement in the supplemental material would be good here? The reason this caught my attention is because a significantly difference rate of change is reported here compared to the Bering Sea estimate of Takahashi et al., 2009. I also note that your Bering Sea data are limited to the most southern region near the Aleutian Island with no grid cells over the shelf.

[R3A2] In the original manuscript, the data used to the full year record were indeed not deseasonalized because of the lack of complete yearly time series to perform such deseasonalization. In the updated manuscript, however, the calculations involving data from the entire year have be deseasonalized using a recently developed data product containing monthly pCO₂ values over the 1998-2015 period over continental shelf waters at 0.25 degree resolutions

(Laruelle et al., 2017). This coastal pCO₂ field is generated using a two-steps artificial neural network interpolation method relying on SOCAT data, similar to that used by (Landschützer et al., 2013) for the global open ocean. The details of the procedure are provided in the Supplementary Material (Lines 80-85).

“The calculation of the year-long trends was performed using all available data from any month over the studied period (1980-2014). A monthly pCO₂ climatology for continental shelf seas calculated using artificial neuronal network interpolations from the SOCAT database (Laruelle et al., 2017) was used to establish an average seasonal pCO₂ cycle for each grid cell. This signal was then removed from the raw data to perform a deseasonalization prior to calculate the linear regressions.”

You reference the Oregon shelf as one of the 15 regions, but Figure 2F isn't actually showing a majority of Oregon data. Only the “wide” definition data are actually off the Oregon coast, the “narrow” data are WA and BC. This area, specifically the data dense region near the Strait of Juan de Fuca, has been examined somewhat in the following paper:

Evans, W., B. Hales, P. G. Strutton, and D. Ianson (2012), Sea-air CO₂ fluxes in the western Canadian coastal ocean, Progress in Oceanography, doi: 10.1016/j.pocean.2012.1001.1003.

[R3A3] Following the reviewer's remark, the region will be renamed more accurately 'NorthWest US coast'. Also, a reference to the study suggested by the reviewer has been introduced into the manuscript when describing the results for this area.

It may be the size of the squares for the “narrow” definition, but the look to go past the 100 km from shore “wide” delineation. At 50N, 0.5 degrees of longitude is 36 km, and I count 6 squares along the lowest band of cells on the WA state cluster.

[R3A4] The reviewer is correct and some of the cells classified are part of the 'narrow shelf' are indeed located further than 100km away from the coastline but they are characterized by depth shallower than 200m and thus belong to the narrow shelf according to our definition. In addition, the minimum distance to the land should be calculated diagonally in this instance and does not exceed 4 squares (~100-150km at this latitude).

It may also be helpful to match the color scale of Figure 1, with yellow centered on zero but with more gradation. There are light green areas in Figure 3 that were yellow in Figure 1.

[R3A5] In the updated manuscript, we modified the colours of the dots on figure 1 to match the colour scale of figure 2 as requested by the reviewer.

References

- Bakker, D. C. E. et al. An update to the Surface Ocean CO₂ Atlas (SOCAT version 2). *Earth Syst. Sci. Data* 6, 69-90, doi:10.5194/essd-6-69-2014 (2014).
- Bauer, J. E. et al. The changing carbon cycle of the coastal ocean. *Nature* 504, 61-70, doi:10.1038/nature12857 (2013).
- Borges A.V. (2005) Do we have enough pieces of the jigsaw to integrate CO₂ fluxes in the Coastal Ocean ? *Estuaries*, 28(1): 3-27
- Borges, A. V., Delille, B. & Frankignoulle, M. Budgeting sinks and sources of CO₂ in the coastal oceans: Diversity of ecosystems counts. *Geophys. Res. Lett.* 32, L14601, doi:10.1029/2005GL023053 (2005).
- Cai, W.-J. Estuarine and coastal ocean carbon paradox: CO₂ sinks or sites of terrestrial carbon incineration? *Annual Review of Marine Science* 3, 123-145, doi:10.1146/annurev-marine-120709-142723 (2011).
- Chen C.T.A. & A.V. Borges (2009) Reconciling opposing views on carbon cycling in the coastal ocean: continental shelves as sinks and near-shore ecosystems as sources of atmospheric CO₂, *Deep-Sea Research II*, 56 (8-10), 578-590.
- Chou, W.-C., et al. The carbonate system in the East China Sea in winter. *Marine Chemistry* 123, 44-55 (2011).
- Hauri et al. Two decades of inorganic carbon dynamics along the West Antarctic Peninsula. *Biogeosciences*, 12, 6761–6779, doi: 10.5194/bg-12-6761-2015 (2015).
- Landschützer, P. et al. A neural network-based estimate of the seasonal to inter-annual variability of the Atlantic Ocean carbon sink, *Biogeosciences*, 10, 7793-7815, doi:10.5194/bg-10-7793-2013 (2013).
- Laruelle, G. G., et al. Evaluation of sinks and sources of CO₂ in the global coastal ocean using a spatially-explicit typology of estuaries and continental shelves, *Geophys. Res. Lett.*, 37, L15607, doi: 10.1029/2010gl043691 (2010.)
- Laruelle, G. G., et al. Global high resolution monthly pCO₂ climatology for the coastal ocean derived from neural network interpolation, *Biogeosciences Discuss.*, <https://doi.org/10.5194/bg-2017-64>, accepted, 2017.
- Le Quéré, C., et al. Impact of climate change and variability on the global oceanic sink of CO₂, *Global Biogeochem. Cycles*, 24, GB4007, doi:10.1029/2009GB003599 (2010)
- Mackenzie, F. T. , Lerman, A., & DeCarlo, E. H. Coupled C, N, P, and O biogeochemical cycling at the land-ocean interface. In: *Treatise on Ocean and Estuarine Science*, v. 4, J Middleburg and R. Laane, eds., Elsevier, 317-342 (2011).
- McNeil, B. I., & Sasse, T. P. Future ocean hypercapnia driven by anthropogenic amplification of the natural CO₂ cycle. *Nature*, 529, 383-386, doi:10.1038/nature16156 (2016)

Salt, L. A., et al. Variability of North Sea pH and CO₂ in response to North Atlantic Oscillation forcing. *J. Geophys. Res. Biogeosci.*, 118, 1584-1592, doi:10.1002/2013JG002306 (2013).

Takahashi, T., et al. Decadal variation of the surface water pCO₂ in the western and central Equatorial Pacific. *Science* 302, 852-856 (2003)

Takahashi, T. et al. Decadal change of the surface water pCO₂ in the North Pacific: a synthesis of 35 years of observations. *J. Geophys. Res.* 111, C07S05 (2006).

Takahashi, T. et al. Climatological mean and decadal change in surface ocean pCO₂, and net sea-air CO₂ flux over the global oceans. *Deep Sea Research Part II: Topical Studies in Oceanography* 56, 554-577, doi:10.1016/j.dsr2.2008.12.009 (2009).

Walsh, J. J. *On the Nature of Continental Shelves*. Academic Press, (1988)

REVIEWERS' COMMENTS:

Reviewer #1 (Remarks to the Author):

The authors have made a clear effort to take on board their review and clarify their work. They have taken a great deal of care to systematically address all points of the review, and the article is improved beyond its originally good standard. Specifically, the authors have clarified their use of wintertime data by including an analysis of full year data, as well as added further comments highlighting the distinction between the wintertime trends they report and annual trends that remain poorly determined. Furthermore, the expanded account of their statistical methods strengthens the work. The article presents a useful step towards an improved understanding of changing biogeochemistry on continental shelves by establishing what the limited extant data show, while also pointing out the need for improvements in shelf data coverage.

Minor comments:

The authors have helpfully added indications of subpanels to their figures (such as “Fig. 1A” Line 175, in the captions etc.) but the figures themselves do not yet appear to have labels on the subpanels.

Line specific points:

73: “towards atmospheric value” should perhaps be “towards the atmospheric value” or “towards atmospheric values”

178: The sentence ending “all year long³⁷” appears to be missing a full stop.

Reviewer #2 (Remarks to the Author):

Using the SOCAT database, the authors calculate linear regressions in 825 cells of half degree resolution. The 825 cells represent only 6% of the global continental shelves. Within this 6%

coverage, only 325 cells out of 825 present a significant increase of the air-sea pCO₂ gradient over time. Figure 2 shows that most of the shelves are located in the temperate latitudes of the northern hemisphere so that their analysis is biased towards these latitudes. The continental shelves show differences between latitude bands. High latitude and temperature continental shelves are sinks of CO₂ while tropical and subtropical shelves are sources of CO₂ (e.g. Chen and Borges 2009).

□pCO₂

However, the authors conclude that the continental shelves are an increasing sink of atmospheric CO₂. With such a small coverage and lack of representativeness, it is wrong to extrapolate to the global continental shelves.

The revised version has toned down some of the statements. However, the authors still draw a conclusion regarding the global trend although they could only determine the trend at a few locations. As such, the paper is still misleading and oversell the results. The title “continental shelves as a variable but increasing sink for atmospheric carbon dioxide” suggests a global result.

The last part of the paper entitled implications for the global carbon budget, as noted by the authors, is highly speculative as they assume that the linear trends apply for the period 1950-2010.

The authors should present the trends only where they could be determined (cf figure 4a in LeQuéré et al., 2010) and explain the mechanisms responsible for the trends there.

Reviewer #3 (Remarks to the Author):

Title: Continental shelves as a variable but increasing sink for atmospheric carbon dioxide

Authors: Laruelle et al

Reviewer: Wiley Evans

Review: The revised manuscript by Laruelle et al addressed each of my comments on the previous draft and, in my opinion, the comments from the other reviewers. The results presented are important, certainly Nature worthy, and ready to publish. My only remaining comment is that the revised term "Northwest US" includes data collected in Canadian waters. An alternative could be "Cascadian continental shelf".

REVIEWERS' COMMENTS:

Reviewer #1 (Remarks to the Author):

The authors have made a clear effort to take on board their review and clarify their work. They have taken a great deal of care to systematically address all points of the review, and the article is improved beyond its originally good standard. Specifically, the authors have clarified their use of wintertime data by including an analysis of full year data, as well as added further comments highlighting the distinction between the wintertime trends they report and annual trends that remain poorly determined. Furthermore, the expanded account of their statistical methods strengthens the work. The article presents a useful step towards an improved understanding of changing biogeochemistry on continental shelves by establishing what the limited extant data show, while also pointing out the need for improvements in shelf data coverage.

We thank the referee for his/her positive evaluation of the revisions we conducted on the manuscript.

Minor comments:

The authors have helpfully added indications of subpanels to their figures (such as “Fig. 1A” Line 175, in the captions etc.) but the figures themselves do not yet appear to have labels on the subpanels.

All panels now contain a letter to facilitate their identification. Note that additional modifications have been implemented to comply with the journal editorial guidelines.

Line specific points:

73: “towards atmospheric value” should perhaps be “towards the atmospheric value” or “towards atmospheric values”

Following the reviewer’s suggestion, the sentence now reads: ‘Data from two large semi-enclosed shelf seas (North Sea and Baltic Sea) and from the Bering Sea suggest that continental shelves may exhibit a rapid increase in $p\text{CO}_2^{22,23}$ towards atmospheric values, ...’

178: The sentence ending “all year long³⁷” appears to be missing a full stop.

Indeed, a full stop was missing after the reference (now reference 32) and has been added.

Reviewer #2 (Remarks to the Author):

Using the SOCAT database, the authors calculate linear regressions in 825 cells of half degree resolution. The 825 cells represent only 6% of the global continental shelves. Within this 6% coverage, only 325 cells out of 825 present a significant increase of the air-sea $p\text{CO}_2$ gradient $\Delta p\text{CO}_2$ over time.

It is true that the 825 cells for which trends in $\Delta p\text{CO}_2$ over time could be calculated in winter only cover 6% of the continental shelves. However, it should be noted that this number stems from the high spatial resolution used in our analysis (0.5 degrees) and, until recently (Laruelle et al., 2014), all global studies investigating the dynamics of the air-water CO_2 exchange in continental shelves have been based on much smaller data coverage (e.g., Borges et al., 2005, Chen and Borges, 2009, Laruelle et al., 2010, Cai, 2011; Chen et al., 2013). Moreover, the reviewer's statement that '*only 325 cells out of 825 present a significant increase of the air-sea $p\text{CO}_2$ gradient $\Delta p\text{CO}_2$ over time*' is incorrect. Indeed, 325 cells correspond to the number of cells for which an increase in $\Delta p\text{CO}_2$ over time faster than $0.5 \mu\text{atm yr}^{-1}$ could be calculated with $p < 0.05$. If anything, this number of cells should thus be compared either to the number of cells for which any trend could be calculated with a significance at $p < 0.05$ (374) or to the number of cells in which the increase in $\Delta p\text{CO}_2$ is faster than $0.5 \mu\text{atm yr}^{-1}$ (495).

Figure 2 shows that most of the shelves are located in the temperate latitudes of the northern hemisphere so that their analysis is biased towards these latitudes. The continental shelves show differences between latitude bands. High latitude and temperature continental shelves are sinks of CO_2 while tropical and subtropical shelves are sources of CO_2 (e.g. Chen and Borges 2009).

The latitudinal distribution of the air-water CO_2 exchange the reviewer is referring to is indeed well known and was already suggested in Borges et al. (2005) and Cai et al. (2006). It is correct that, particularly at low latitudes, continental shelf waters may be characterized by $p\text{CO}_2$ higher than that of the atmosphere and may thus behave as atmospheric sources of CO_2 . This fact is clearly mentioned in our text and additional emphasis has now been added in the updated manuscript:

"Despite great local variability, the data also suggest that mid- to high- latitude shelves are generally a sink of CO_2 , while warm tropical shelves are a moderate source of CO_2 ^{14,16,17}. A broad consensus regarding the current strength of the global shelf CO_2 sink and its large-scale spatial variability has thus recently emerged. In particular, continuous high resolution $p\text{CO}_2$ maps for continental shelf seas derived from the interpolation of experimental data¹⁹ clearly support this latitudinal trend in all oceanic basins."

However, the authors conclude that the continental shelves are an increasing sink of atmospheric CO_2 . With such a small coverage and lack of representativeness, it is wrong to extrapolate to the global continental shelves.

We understand the reviewer's concern regarding the representativeness of the 825 cells we use as basis to derive global conclusions regarding the temporal evolution of the intensity of the atmospheric CO_2 uptake by continental shelves. Although these cells only account for a fraction of the global continental shelf surface, they are widely distributed around the world and allow the clear identification of regional trends in 15 different areas. It is true that a bias towards temperate latitudes exists (as explicitly mentioned in the manuscript) but this bias reflects the current sampling distribution in global databases such as SOCAT. The comparison of the distribution of our cells with typologies of continental shelves such as Liu's classification (Liu et al., 2010) or the MARCATS segmentation (Laruelle et al., 2013) confirms the under-representation of tropical shelves in our analysis and highlights the lack of sampling in regions under Monsoonal influence.

However, polar, sub-polar, temperate, eastern and western boundary currents are all represented as supported by their respective data availability.

In order to better evaluate how sampling bias potentially limits the representativeness of our analysis, we examined the distribution of our 825 cells in the parameter space of average winter $p\text{CO}_2$ as a function of SST (Supplementary Figure 1)

Supplementary figure 1: Distribution of pairs of winter $p\text{CO}_2$ and SST for all cells extracted from the SOM_FFNet data product of Laruelle et al. (2017) (black dot), the SOCAT database (blue dots) and the 825 cells for which trends in $d\Delta p\text{CO}_2/dt$ were calculated in this study (red dots).

The distribution of the cells investigated in our study intersect a large fraction of all possible combinations of winter $p\text{CO}_2$ and SST conditions for continental shelf waters according to the data produced by the artificial neural network interpolation produced by Laruelle et al (2017). Therefore, we consider it as a fairly representative sample of the global distribution, such that a global extrapolation is defensible. At the same time, it is clear that the distribution is not unbiased. Most striking is the under-representation of coastal areas with high $p\text{CO}_2$ and SST condition in the low latitude, which is a direct consequence of a serious undersampling of these regions in the SOCAT data base (Bakker et al., 2016, blue). If these regions exhibit as a whole an entirely different trend from the other regions, then our global extrapolation can indeed be biased. At present, we do not have the data to demonstrate this, but Figure 2, where the comparison between the distributions between the cells for which $d\Delta p\text{CO}_2/dt > 0$ and those for which $d\Delta p\text{CO}_2/dt < 0$ reveals very little difference, suggest that the trends are not clustered but relatively randomly distributed.

New supplementary figure 2: Distribution of pairs of winter $p\text{CO}_2$ and SST for cells with $d\Delta p\text{CO}_2/dt > 0$ (red, left) and $d\Delta p\text{CO}_2/dt < 0$ (blue, right).

The revised version has toned down some of the statements. However, the authors still draw a conclusion regarding the global trend although they could only determine the trend at a few locations. As such, the paper is still misleading and oversell the results. The title “continental shelves as a variable but increasing sink for atmospheric carbon dioxide” suggests a global result.

For the reasons exposed in our answers above, we still consider it statistically justifiable to use the arguably limited data to arrive at a global extrapolation. We fully agree that great care must be taken in this step, that the caveats need to be clearly stated, and also that the data are not over interpreted. To this end, we have modified some of the text in the corresponding sections:

“Globally, our analysis of the 825 temporal trends in $\Delta p\text{CO}_2$ using winter only data covers a shelf surface area of $1.4 \times 10^6 \text{ km}^2$, which represents $\sim 6\%$ of the global continental shelves. This includes data from the more isolated cells that were not considered in the previous section. While the coverage is relatively small, heterogeneous and somewhat skewed towards temperate latitudes in the northern hemisphere, it nevertheless covers most of the range of $p\text{CO}_2$ and SST encountered in continental shelf waters (Supplementary Figures 1&2). Exceptions are the low latitudes, which are poorly represented in our data set. While we need to recognize this limitation, the broad coverage in terms of environmental conditions permits us to assemble all estimated trends and assume that they represent a sufficiently unbiased sample of the shelf trends across the globe.”

We agree, however, that attention must be paid to the notion of increasing atmospheric CO₂ sink when some continental shelves (especially in the tropics) are actually sources of CO₂. In such case, our calculations suggest that these CO₂ source are becoming less intense over time. Thus, in the updated manuscript, we paid attention not to refer to regions that are source of CO₂ for the atmosphere as increasing CO₂ sinks but rather decreasing CO₂ source. We also modified a potentially misleading sentence (line 675 in the marked up version of the updated manuscript) from

“...show trends that are dominantly in accordance with an increase of the CO₂ sink on continental shelves.”

into

“...show trends that are dominantly in accordance with an increase in ΔpCO₂ over time.”

Last, to avoid any confusion, we propose to add the word ‘global’ to the title of the manuscript to better convey the idea that, globally, continental shelves are a net CO₂ sink but not necessarily everywhere.

The new title reads:

“Continental shelves as a variable but increasing global sink for atmospheric carbon dioxide”

The last part of the paper entitled implications for the global carbon budget, as noted by the authors, is highly speculative as they assume that the linear trends apply for the period 1950-2010.

The authors should present the trends only where they could be determined (cf figure 4a in LeQuéré et al., 2010) and explain the mechanisms responsible for the trends there.

We follow the reviewer’s suggestion in the updated manuscript and supplementary information. We now pay attention to distinguish between the 1988-2007 period for which observations allowed us to derive a trend in $d\Delta pCO_2/dt$ and earlier times. We thus modified the supplementary figure 3 in a way that explicitly differentiates the period for which the trend in $d\Delta pCO_2/dt$ is derived from data (1988-2007, continuous line), from the earlier period during which the trend is extrapolated (dashed line).

We also made an effort to make the text very explicit regarding which part of the trend is derived from data and which part is speculative (i.e. the time at which continental shelf seas might have switch from atmospheric CO₂ sources to atmospheric CO₂ sinks).

“This approach allows comparing trends derived from observations alone with earlier modelling work and will stimulate further exploration of coastal CO₂ trend^{28,30}. Our calculations suggest that the magnitude of the average winter ΔpCO₂ increased by 69% over the 1988-2007 period. By extrapolating this trend to earlier times, it can be speculated that the continental shelves might indeed have turned from a purported pre-industrial source of CO₂ into a sink for atmospheric CO₂ as early as in the 1950s, at least during wintertime (Supplementary Figure 3).”

Reviewer #3 (Remarks to the Author):

Title: Continental shelves as a variable but increasing sink for atmospheric carbon dioxide

Authors: Laruelle et al

Reviewer: Wiley Evans

Review: The revised manuscript by Laruelle et al addressed each of my comments on the previous draft and, in my opinion, the comments from the other reviewers. The results presented are important, certainly Nature worthy, and ready to publish. My only remaining comment is that the revised term "Northwest US" includes data collected in Canadian waters. An alternative could be "Cascadian continental shelf".

We thank the reviewer for his positive comment. We agree that some cells included in the region named 'Northwest US' indeed belong to Canadian territorial waters. We thus followed the reviewer's suggestion and replaced Northwest US shelf by ' Cascadian shelves' throughout the manuscript and in table 1.

References:

Bakker, D. C. E. et al. A multi-decade record of high-quality fCO₂ data in version 3 of the Surface Ocean CO₂ Atlas (SOCAT), *Earth Syst. Sci. Data*, 8, 383-413, doi:10.5194/essd-8-383-2016 (2016).

Borges, A. V., Delille, B. & Frankignoulle, M. Budgeting sinks and sources of CO₂ in the coastal oceans: Diversity of ecosystems counts. *Geophys. Res. Lett.* 32, L14601, doi:10.1029/2005GL023053 (2005).

Cai, W.-J., M. H. Dai, and Y. C. Wang (2006), Air sea exchange of carbon dioxide in ocean margins: A province based synthesis, *Geophys. Res. Lett.*, 33, L12603, doi:10.1029/2006GL026219.

Cai, W.-J. (2011), Estuarine and coastal ocean carbon paradox: CO₂ sinks or sites of terrestrial carbon incineration?, *Ann. Rev. Mar. Sci.*, 3, 123–145, doi:10.1146/annurev-marine-120709-142723.

Chen, C. T. A., and A. V. Borges (2009), Reconciling opposing views on carbon cycling in the coastal ocean: Continental shelves as sinks and near-shore ecosystems as sources of atmospheric CO₂, *Deep Sea Res., Part II*, 56(8–10), 578–590, doi:10.1016/j.dsr2.2009.01.001.

Chen, C. T. A., T. H. Huang, Y. C. Chen, Y. Bai, X. He, and Y. Kang (2013), Air-sea exchanges of CO₂ in the world's coastal seas, *Biogeosciences*, 10, 6509–6544, doi:10.5194/bg-10-6509-2013.

Laruelle, G. G. Dürr, H. H., Slomp, C. P., & Borges, A. V. Evaluation of sinks and sources of CO₂ in the global coastal ocean using a spatially-explicit typology of estuaries and continental shelves, *Geophys. Res. Lett.*, 37, L15607, doi:10.1029/2010GL043691 (2010).

Laruelle, G. G. et al. Global multi-scale segmentation of continental and coastal waters from the watersheds to the continental margins, *Hydrol. Earth Syst. Sci.*, **17**, 2029–2051, doi:10.5194/hess-17-2029-2013 (2013).

Laruelle, G. G., Landschützer, P., Gruber, N., Tison, J.-L., Delille, B., & Regnier, P. Global high resolution monthly pCO₂ climatology for the coastal ocean derived from neural network interpolation, *Biogeosciences*, **14**, 4545-4561, doi:10.5194/bg-14-4545-2017 (2017).

Laruelle, G. G., Lauerwald, R., Pfeil, B. & Regnier, P. Regionalized global budget of the CO₂ exchange at the air-water interface in continental shelf seas. *Global Biogeochemical Cycles* **28**, 2014GB004832, doi:10.1002/2014GB004832 (2014).

Liu, K.-K., L. Atkinson, R. Quinones, and L. Talaue-McManus (2010), Carbon and Nutrient Fluxes in Continental Margins, *Global Change—The IGBP Series*, vol. 3, edited by K.-K. Liu et al., Springer, Berlin, Heidelberg.